# On the Importance of Exploration for Generalization in Reinforcement Learning

**Yiding Jiang**[*]
Carnegie Mellon University
yidingji@cs.cmu.edu

**J. Zico Kolter**
Carnegie Mellon University
zkolter@cs.cmu.edu

**Roberta Raileanu**
Meta AI Research
raileanu@meta.com

## Abstract

Existing approaches for improving generalization in deep reinforcement learning (RL) have mostly focused on representation learning, neglecting RL-specific aspects such as exploration. We hypothesize that the agent's exploration strategy plays a key role in its ability to generalize to new environments. Through a series of experiments in a tabular contextual MDP, we show that exploration is helpful not only for efficiently finding the optimal policy for the training environments but also for acquiring knowledge that helps decision making in unseen environments. Based on these observations, we propose EDE: Exploration via Distributional Ensemble, a method that encourages the exploration of states with high epistemic uncertainty through an ensemble of Q-value distributions. The proposed algorithm is the first value-based approach to achieve strong performance on both Procgen and Crafter, two benchmarks for generalization in RL with high-dimensional observations. The open-sourced implementation can be found at https://github.com/facebookresearch/ede.

## 1   Introduction

Current deep reinforcement learning (RL) algorithms struggle to generalize in *contextual* MDPs (CMDPs) where agents are trained on a number of different environments that share a common structure and tested on unseen environments from the same family [22, 112, 75], despite being competitive in *singleton* Markov decision processes (MDPs) where agents are trained and tested on the same environment [73, 43, 8]. This is particularly true for value-based methods [113] (*i.e.,* methods that directly derive a policy from the value functions), where there has been little progress on generalization relative to policy-optimization methods (*i.e.,* methods that learn a parameterized policy in addition to a value function) [26]. Most existing approaches for improving generalization in CMDPs have treated this challenge as a pure representation learning problem, applying regularization techniques which are commonly used in supervised deep learning [29, 21, 45, 61, 116, 60, 92]. However, these methods neglect the unique structure of reinforcement learning (RL), namely that agents collect their own data by exploring their environments. This suggests that there may be other avenues for improving generalization in RL beyond representation learning.

> Exploration can help an agent gather more information about its environment, which can improve its ability to generalize to new tasks or environments.

While this statement is seemingly intuitive, a formal and explicit connection has surprisingly not been made outside of more niche sub-areas of RL (*e.g.,* task-agnostic RL [123] or meta-RL [65]), nor has there been a convincing empirical demonstration of this statement on common generalization

---

[*]Work done while interning at Meta AI Research.

37th Conference on Neural Information Processing Systems (NeurIPS 2023).

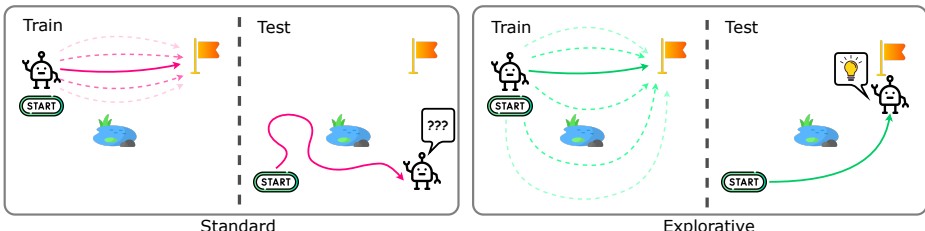

Figure 1: Exploration can help agents learn about parts of the environment which may be useful at test time, even if they are not needed for the optimal policy on the training environments. Note that this picture only illustrates one of many scenarios where exploration helps.

benchmarks for RL. In this work, we show that the agent's exploration strategy is a key factor influencing generalization in contextual MDPs. First, exploration can accelerate training in deep RL, and since neural networks tend to naturally generalize, better exploration can result in better training performance and consequently better test performance. More interestingly, in singleton MDPs, exploration can only benefit decisions in that environment, while in CMDPs exploration in one environment can also help decisions in other, potentially unseen, environments. This is because learning about different parts of the environment can be useful in other MDPs even if it is not useful for the current MDP. As shown in Figure 1, trajectories that are suboptimal in certain MDPs may turn out to be optimal in other MDPs from the same family, so this knowledge can help find the optimal policy more quickly in other MDPs encountered during training, and better generalize to new MDPs without additional training.

One goal of exploration is to learn new information about the (knowable parts of the) environment so as to reduce *epistemic uncertainty*. To model epistemic uncertainty (which is reducible by acquiring more data), we need to disentangle it from aleatoric uncertainty (which is irreducible and stems from the inherent stochasticity of the environment). As first observed by Raileanu and Fergus [90], in CMDPs the same state can have different values depending on the environment, but the agent does not know which environment it is in so it cannot perfectly predict the value of such states. This is a type of aleatoric uncertainty that can be modeled by learning a distribution over possible values rather than a single point estimate [10]. Based on these observations, we propose *Exploration via Distributional Ensemble* (EDE), a method that uses an ensemble of Q-value distributions to encourage exploring states with large epistemic uncertainty. We evaluate EDE on both Procgen [22] and Crafter [37], two procedurally generated CMDP benchmarks for generalization in deep RL, demonstrating a significant improvement over more naive exploration strategies. This is the first model-free value-based method to achieve state-of-the-art performance on these benchmarks, in terms of both sample efficiency and generalization, surpassing strong policy-optimization baselines and even a model-based one. Crucially, EDE only targets exploration so it can serve as a strong starting point for future works.

To summarize, our work makes the following contributions: (i) identifies exploration as a key factor for generalization in CMDPs and supports this hypothesis using a didactic example in a tabular CMDP, (ii) proposes an exploration method based on minimizing the agent's epistemic uncertainty in high-dimensional CMDPs, and (iii) achieves state-of-the-art performance on two generalization benchmarks for deep RL, Procgen and Crafter.

## 2   Background

**Episodic Reinforcement Learning.** A *Markov decision process* (MDP) is defined by the tuple $\mathcal{M} = (\mathcal{S}, \mathcal{A}, R, P, \gamma, \mu)$, where $\mathcal{S}$ is the state space, $\mathcal{A}$ is the action space, $R : \mathcal{S} \times \mathcal{A} \to [R_{\min}, R_{\max}]$ is the reward function , $P : \mathcal{S} \times \mathcal{A} \times \mathcal{S} \to \mathbb{R}_{\geq 0}$ is the transition distribution, $\gamma \in (0, 1]$ is the discount factor, and $\mu : \mathcal{S} \to \mathbb{R}_{\geq 0}$ is the initial state distribution. We further denote the trajectory of an episode to be the sequence $\tau = (s_0, a_0, r_0, \ldots, s_T, a_T, r_T, s_{T+1})$ where $r_t = R(s_t, a_t)$ and $T$ is the length of the trajectory which can be infinite. If a trajectory is generated by a probabilistic policy $\pi : \mathcal{S} \times \mathcal{A} \to \mathbb{R}_{\geq 0}$, $Z^\pi = \sum_{t=0}^{T} \gamma^t r_t$ is a random variable that describes the *discounted return* the policy achieves. The objective is to find a $\pi^\star$ that maximizes the expected discounted return, $\pi^\star = \arg\max_\pi \mathbb{E}_{\tau \sim p^\pi(\cdot)} [Z^\pi]$, where $p^\pi(\tau) = \mu(s_0) \prod_{t=0}^{T} P(s_{t+1} \mid s_t, a_t) \pi(s_t \mid a_t)$. For simplicity, we will use $\mathbb{E}_\pi$ instead of $\mathbb{E}_{\tau \sim p^\pi(\cdot)}$ to denote the expectation over trajectories sampled

from the policy $\pi$. With a slight abuse of notation, we use $Z^\pi(s, a)$ to denote the conditional discounted return when starting at $s$ and taking action $a$ (*i.e.*, $s_0 = s$ and $a_0 = a$). Finally, without loss of generality, we assume all measures are discrete and their values lie within $[0, 1]$.

**Value-based methods** [113] rely on a fundamental quantity in RL, the state-action value function, also referred to as the *Q-function*, $Q^\pi(s, a) = \mathbb{E}_\pi [Z^\pi \mid s_0 = s, \ a_0 = a]$. The Q-function of a policy can be found at the fixed point of the Bellman operator, $\mathcal{T}^\pi$ [11], $\mathcal{T}^\pi Q(s, a) = \mathbb{E}_{s' \sim P(\cdot|s,a), \ a' \sim \pi(\cdot|s')} [R(s, a) + \gamma Q(s', a')]$. The *value function* for a state $s$ is defined as $V^\pi(s) = \mathbb{E}_{a \sim \pi(\cdot|s)}[Q^\pi(s, a)]$. Bellemare et al. [10] extends the procedure to the distribution of discounted returns, $\mathcal{T}^\pi Z(s, a) \overset{d}{=} R(s, a) + \gamma Z(s', a'), \quad s' \sim P(\cdot \mid s, a)$ and $a' \sim \pi(\cdot \mid s')$, where $\overset{d}{=}$ denotes that two random variables have the same distributions. This extension is referred to as *distributional RL* (we provide a more detailed description of QR-DQN, the distributional RL algorithm we use, in Appendix D.1). For value-based methods, the greedy policy is directly derived from the Q-function as $\pi(a \mid s) = \mathbb{1}_{\arg\max_{a'} Q(a', s)}(a)$.

**Generalization in Contextual MDPs.** A *contextual Markov decision process* (CMDP) [39] is a special class of *partially observable Markov decision process* (POMDP) consisting of different MDPs, that share state and action spaces but have different $R$, $P$, and $\mu$. In addition to the standard assumptions of a CMDP, we assume the existence of a structured distribution $\Psi$ over the MDPs. During training, we are given a (finite or infinite) number of training MDPs, $\tilde{\mathcal{M}}_{\text{train}} = \{\mathcal{M}_1, \mathcal{M}_2, \ldots, \mathcal{M}_n\}$, drawn from $\Psi$ (Ghosh et al. [35] refers to this setting as epistemic POMDP). For each $\mathcal{M}$, we use $Q^\pi_\mathcal{M}(s, a)$ and $V^\pi_\mathcal{M}(s)$ to denote the Q-function and value function of that $\mathcal{M}$. We use $p^{\pi,\mathcal{M}}(\tau)$ to denote the trajectory distribution of rolling out $\pi$ in $\mathcal{M}$. The objective is to find a single policy $\pi$ that maximizes the expected discounted return over the entire distribution of MDPs, $\mathbb{E}_{\tau \sim p^{\pi,\mathcal{M}}(\cdot), \mathcal{M} \sim \Psi(\cdot)} \left[ \sum_{t=0}^T \gamma^t r_t \right]$ without retraining on the unseen $\mathcal{M}$ (*i.e.,* zero-shot generalization). We will refer to this quantity as the *test return*[2].

Since in RL the algorithm collects its own data, the appropriate notion of generalization is the performance a learning algorithm can achieve given a finite number of interactions. Crucially, if algorithm 1 achieves a better test return than algorithm 2 given the same number of interactions with $\mathcal{M}_{\text{train}}$, we can say that algorithm 1 generalizes better than algorithm 2. Furthermore, we assume the existence of $\pi^\star$ (potentially more than one) that is $\alpha$-optimal for all $\mathcal{M} \in \text{supp}(\Psi)$ and all $s_0$ in $\text{supp}(\mu_\mathcal{M})$, *i.e.,* $V^{\pi^\star}_\mathcal{M}(s_0) \geq \max_\pi V^\pi_\mathcal{M}(s_0) - \alpha$ where $\alpha$ is small. This assumption only ensures that zero-shot generalization is intractable [69] but it is usually not used explicitly in algorithm design. If the number of training environments is infinite, the challenge is learning good policies for all of them in a sample-efficient manner, *i.e.,* optimization; if it is finite, the challenge is also generalization to unseen environments.

# 3 Generalization in a Tabular CMDP

Much of the literature treats generalization in deep RL as a pure representation learning problem [100, 119, 120, 1] and aims to improve it by using regularization [29, 118, 21, 45] or data augmentation [21, 61, 116, 56, 60, 92, 112]. In this section, we will first show that the problem of generalization in RL extends **beyond representation learning** by considering a tabular CMDP, which does not require any representation learning. The goal is to provide intuition on the role of exploration for generalization in RL using a toy example. More specifically, we show that exploring the training environments can be helpful not only for finding rewards in those environments but also for making good decisions in new environments encountered at test time.

Concretely, we consider a generic $5 \times 5$ grid environment (Figure 2a). During training, the agent always starts at a fixed initial state, $(x = 0, y = 0)$, and can move in 4 cardinal directions (*i.e.,* up, down, left, right). The transition function is deterministic and if the agent moves against the boundary, it ends up at the same location. If the agent reaches the terminal state, $(x = 4, y = 0)$, it receives a large positive reward, $r = 2$ and the episode ends. Otherwise, the agent receives a small negative

---

[2]It may be useful to contrast this with the notion of generalization gap, the difference between training and test returns. Without good training performance, the generalization gap is not a meaningful measure: a random policy would have no generalization gap at all. The generalization gap is important for supervised learning because we can get nearly perfect training performance, but this is not the case for RL.

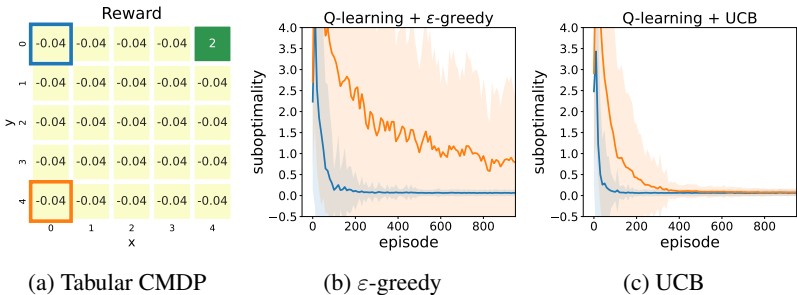

(a) Tabular CMDP  (b) $\varepsilon$-greedy  (c) UCB

Figure 2: (a) A tabular CMDP that illustrates the importance of exploration for generalization in RL. During training, the agent starts in the **blue** square, while at test time it starts in the **orange** square. In both cases, the goal is to get to the **green** square. The other plots show the mean and standard deviation of the train and test suboptimality (difference between optimal return and achieved return) over 100 runs for (b) Q-learning with $\varepsilon$-greedy exploration, (c) Q-learning with UCB exploration.

reward, $r = -0.04$. At test time, the agent starts at a *different* location, $(x = 0, y = 4)$. In other words, the train and test MDPs only differ by their initial state distribution. In addition, each episode is terminated at 250 steps (10 times the size of the state space) to speed up the simulation, but most episodes reach the terminal state before forced termination.

We study two classes of algorithms with different exploration strategies: (1) Q-learning with $\varepsilon$-greedy (`Greedy`, Watkins and Dayan [113]), (2) Q-learning with UCB (`UCB`, Auer et al. [6]). To avoid any confounding effects of function approximation, we use *tabular* policy parameterization for Q-values. Both `Greedy` and `UCB` use the same base Q-learning algorithm [113]. `Greedy` explores with $\varepsilon$-greedy strategy which takes a random action with probability $\varepsilon$ and the best action according to the Q-function $\arg\max_a Q(s, a)$ with probability $1 - \varepsilon$. In contrast, `UCB` is uncertainty-driven so it explores according to $\pi(a \mid s) = \mathbb{1}(a = \arg\max_{a'} Q(s, a') + c\sqrt{\log(t)/N(s, a')})$, where $t$ is the total number of timesteps, $c$ is the exploration coefficient, and $N(s, a)$ is the number of times the agent has taken action $a$ in state $s$, with ties broken randomly[3]. While `Greedy` is a naive but widely used exploration strategy [105], `UCB` is an effective algorithm designed for multi-armed bandits [6, 5]. Chen et al. [19] showed that uncertainty-based exploration bonus also performs well in challenging RL environments. See Appendix C for more details about the experimental setup.

Each method's exploration strategy is controlled by a single hyperparamter. For each hyperparameter, we search over 10 values and run every value for 100 trials. Each trial lasts for 1000 episodes. The results (mean and standard deviation) of hyperparameters with the highest average test returns for each method are shown in Figures 2b and 2c. We measure the performance of each method by its *suboptimality*, *i.e.*, the difference between the undiscounted return achieved by the learned policy and the undiscounted return of the optimal policy. While all three methods are able to quickly achieve an optimal return for the training MDP, their performances on the test MDP differ drastically. First, we observe that `Greedy` has the worst generalization performance and the highest variance. On the other hand, `UCB` can reliably find the optimal policy for the test MDP as the final return has a negligible variance[4]. In Appendix C.4, we provide another set of simulations based on *changing dynamics* and make similar observations. These experiments show that **more effective exploration of the training environments can result in better generalization to new environments**.

It is natural to ask whether this example is relevant for realistic deep RL problems. Analytically, we can motivate the importance of exploration for generalization using *sub-MDPs* [51] and the *sample complexity* of Q-learning [63]. Due to space constraints, we develop this argument in Appendix A. Conceptually, the tabular CMDP with two initial state distributions captures a common phenomenon in more challenging CMDPs like Procgen, namely that at test time, the agent can often end up in

---

[3]This `UCB` is a simple extension of the widely used bandits algorithm to MDP and does not enjoy the same regret guarantee, but we find it to be effective for the didactic purpose. Azar et al. [7] presents a formal but much more complicated extension of the UCB algorithm for value iteration on MDPs. The `UCB` used is more similar to a count-based exploration bonus.

[4]While the training and test MDPs are seemingly drawn from different distributions, the setting can be made IID by taking a mixture of the train and test MDP. Since there is a significant discrepancy between train and test performance, all the observations would remain valid.

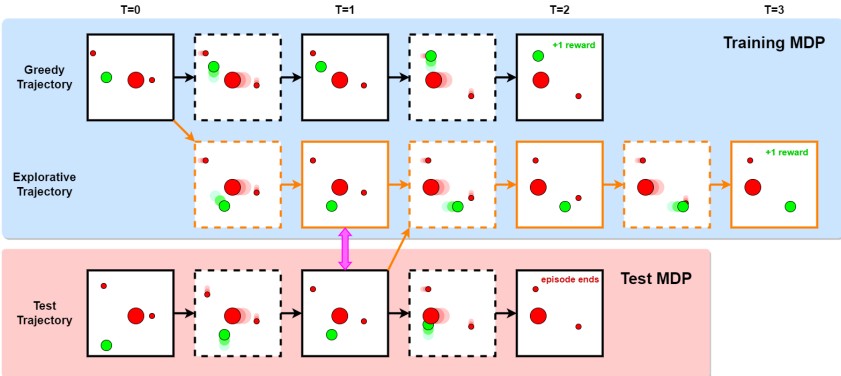

Figure 3: A simplified version of `bigfish` in Procgen (Figure 6) that captures the spirit of the grid world example from Figure 2a. The green dot represents the agent and the red dots represent the enemies (see Appendix B for more details). The goal is to eat smaller dots while avoiding larger dots. An agent that explores both trajectories in the training MDP could recover the optimal behavior on the test MDP whereas an agent that only focuses on solving the training MDP would fail.

suboptimal states that are rarely visited by simple exploration on the training MDPs. Having explored such states during training can help the agent recover from suboptimal states at test time. See Figure 3 and Appendix B for an illustrative example inspired by one of the Procgen games. This is similar to covariate shift in imitation learning where the agent's suboptimality can compound over time. The effect of efficient exploration is, in spirit, similar to that of DAgger [94] — it helps the agent learn how to recover from situations that are rare during training.

In Appendix G.1, we explore some other interpretations for why exploration improves generalization (which are also related to the sample complexity of Q-learning). In addition, we also explore other potential reasons for why current RL methods generalize poorly in Appendix G.2. Note that we do not claim exploration to be the only way of improving generalization in RL as different environments may have very different properties that require distinct generalization strategies. In Appendix C.2 and C.3, we study other potential avenues for improving generalization. The benefits of exploration are complementary to these approaches, and they may be combined to further improve performance. In this paper, we focus on using exploration to improve generalization in contextual MDPs.

## 4   Exploration via Distributional Ensemble

In the previous section, we showed we can improve generalization via exploration in the tabular setting. In this section, we would like to extend this idea to deep RL with function approximation. While in the tabular setting shown above there is no intrinsic stochasticity, environments can in general be stochastic (*e.g.,* random transitions, or random unobserved contexts). At a high level, epistemic uncertainty reflects a lack of knowledge which can be addressed by collecting more data, while aleatoric uncertainty reflects the intrinsic noise in the data which cannot be reduced regardless of how much data is collected. One goal of exploration is to gather information about states with high epistemic uncertainty [83] since aleatoric uncertainty cannot be reduced, but typical estimates can contain both types of uncertainties [52]. In CMDPs, this is particularly important because a large source of aleatoric uncertainty is not knowing which context the agent is in [90].

In this section, we introduce *Exploration via Distributional Ensemble* (EDE), a method that encourages the exploration of states with high epistemic uncertainty. Our method builds on several important ideas from prior works, the most important of which are deep ensembles and distributional RL. While ensembles are a useful way of measuring uncertainty in neural networks [59], such estimates typically contain both epistemic and aleatoric uncertainty. To address this problem, we build on Clements et al. [20] which introduced an approach for disentangling the epistemic uncertainty from the aleatoric uncertainty of the learned Q-values.

**Uncertainty Estimation.** Clements et al. [20] showed that learning the quantiles for QR-DQN [23] can be formulated as a Bayesian inference problem, given access to a posterior $p(\boldsymbol{\theta} \mid \mathcal{D})$, where $\boldsymbol{\theta}$ is the discretized quantiles of $Z(\boldsymbol{s}, \boldsymbol{a})$, and $\mathcal{D}$ is the dataset of experience on which the quantiles are esti-

mated. Let $j \in [N]$ denote the index for the $j^{\text{th}}$ quantile and $\mathcal{U}\{1, N\}$ denote the uniform distribution over integers between 1 and $N$. The uncertainty of the Q-value, $Q(s, a) = \mathbb{E}_{j \sim \mathcal{U}\{1,N\}} [\theta_j(s, a)]$, is the relevant quantity that can inform exploration. The overall uncertainty $\sigma^2 = \text{Var}_{\theta \sim p(\theta|\mathcal{D})} [Q(s, a)]$ can be decomposed into *epistemic uncertainty* $\sigma^2_{\text{epi}}(s, a)$ and *aleatoric uncertainty* $\sigma^2_{\text{ale}}(s, a)$ such that $\sigma^2(s, a) = \sigma^2_{\text{epi}}(s, a) + \sigma^2_{\text{ale}}(s, a)$, where,

$$\sigma^2_{\text{epi}}(s, a) = \mathbb{E}_{j \sim \mathcal{U}\{1,N\}} \left[ \text{Var}_{\theta \sim p(\theta|\mathcal{D})} [\theta_j(s, a)] \right], \ \sigma^2_{\text{ale}}(s, a) = \text{Var}_{j \sim \mathcal{U}\{1,N\}} \left[ \mathbb{E}_{\theta \sim p(\theta|\mathcal{D})} [\theta_j(s, a)] \right]. \quad (1)$$

Ideally, given an infinite or sufficiently large amount of diverse experience, one would expect the posterior to concentrate on the true quantile $\theta^\star$, so $\text{Var}_{\theta \sim p(\theta|\mathcal{D})} [\theta_j(s, a)]$ and consequently $\sigma^2_{\text{epi}}(s, a) = 0$. $\sigma^2_{\text{ale}}$ would be non-zero if the true quantiles have different values. Intuitively, to improve the sample efficiency, the agent should visit state-action pairs with high epistemic uncertainty in order to learn more about the environment [19]. It should be noted that the majority of the literature on uncertainty estimation focuses on supervised learning; in RL, due to various factors such as bootstrapping, non-stationarity, limited model capacity, and approximate sampling, the uncertainty estimation generally contains errors but empirically even biased epistemic uncertainty[5] is beneficial for exploration. We refer interested readers to Charpentier et al. [17] for a more thorough discussion.

Sampling from $p(\theta \mid \mathcal{D})$ is computationally intractable for complex MDPs and function approximators such as neural networks. Clements et al. [20] approximates samples from $p(\theta \mid \mathcal{D})$ with randomized MAP sampling [87] which assumes a Gaussian prior over the model parameters. However, the unimodal nature of a Gaussian in parameter space may not have enough diversity for effective uncertainty estimation [31]. Many works have demonstrated that *deep ensembles* tend to outperform other approximate posterior sampling techniques [59, 31] in supervised learning. Motivated by these observations, we propose to maintain $M$ copies of fully-connected value heads, $g_i : \mathbb{R}^d \rightarrow \mathbb{R}^{|\mathcal{A}| \times N}$ that share a single feature extractor $f : \mathcal{S} \rightarrow \mathbb{R}^d$, similar to Osband et al. [81]. However, we train each value head with *different* mini-batches and random initialization (*i.e.,* deep ensemble) instead of distinct data subsets (*i.e.,* bootstrapping). This is consistent with Lee et al. [62] which shows deep ensembles usually perform better than bootstrapping for estimating the uncertainty of Q-values but, unlike EDE, they do not decompose uncertainties.

Concretely, $i \in [M]$ is the index for $M$ ensemble heads of the Q-network, and $j \in [N]$ is the index of the quantiles. The output of the $i^{\text{th}}$ head for state $s$ and action $a$ is $\theta_i(s, a) \in \mathbb{R}^N$, where the $j^{\text{th}}$ coordinate, and $\theta_{ij}(s, a)$ is the $j^{\text{th}}$ quantile of the predicted state-action value distribution for that head. The finite sample estimates of the two uncertainties are:

$$\hat{\sigma}^2_{\text{epi}}(s, a) = \frac{1}{N \cdot M} \sum_{j=1}^{N} \sum_{i=1}^{M} \left( \theta_{ij}(s, a) - \bar{\theta}_j(s, a) \right)^2, \ \hat{\sigma}^2_{\text{ale}}(s, a) = \frac{1}{N} \sum_{j=1}^{N} \left( \bar{\theta}_j(s, a) - Q(s, a) \right)^2, \quad (2)$$

where $\bar{\theta}_j(s, a) = \frac{1}{M} \sum_{i=1}^{M} \theta_{ij}(s, a)$ and $Q(s, a) = \frac{1}{N} \sum_{j=1}^{N} \bar{\theta}_j(s, a)$.

**Exploration Policy.** There are two natural ways to use this uncertainty. The first one is by using Thompson sampling [110] where the exploration policy is defined by sampling Q-values from the posterior:

$$\pi_{\text{ts}}(a \mid s) = \mathbb{1}_{\arg\max_{a'} \ \xi(s, a')}(a), \quad \text{where} \quad \xi(s, a') \sim \mathcal{N} \left( Q(s, a'), \varphi \ \hat{\sigma}_{\text{epi}}(s, a') \right). \quad (3)$$

$\varphi \in \mathbb{R}_{\geq 0}$ is a coefficient that controls how the agent balances exploration and exploitation. Alternatively, we can use the upper-confidence bound (UCB, Chen et al. [19]):

$$\pi_{\text{ucb}}(a \mid s) = \mathbb{1}_{a^\star}(a), \quad \text{where} \quad a^\star = \arg\max_{a'} Q(s, a') + \varphi \ \hat{\sigma}_{\text{epi}}(s, a'), \quad (4)$$

which we found to achieve better results in CMDPs than Thompson sampling [20] on Procgen when we use multiple parallel workers to collect experience, especially when combined with the next technique.

**Equalized Exploration.** Due to function approximation, the model may lose knowledge of some parts of the state space if it does not see them often enough. Even with UCB, this can still happen after the agent learns a good policy on the training environments. To increase the data diversity, we

---

[5]It may still contain some aleatoric uncertainty due to the heteroskedasticity of the problem.

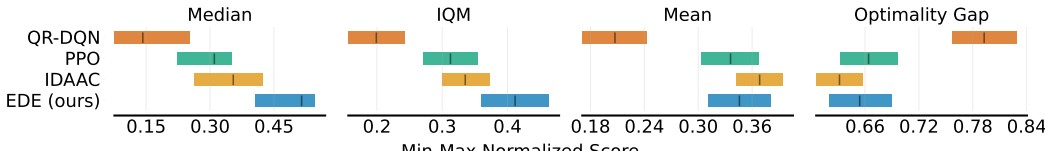

Figure 4: Test performance of different methods on the Procgen benchmark across 5 runs. Our method greatly outperforms all baselines in terms of median and IQM (the more statistically robust metrics) and is competitive with the state-of-the-art policy optimization methods in terms of mean and optimality gap. The optimality gap is equal to 1-mean for the evaluation configuration we chose.

propose to use different exploration coefficients for each copy of the model used to collect experience. Concretely, we have $K$ actors with synchronized weights; the $k^{\text{th}}$ actor collects experience with the following policy:

$$\pi_{\text{ucb}}^{(k)}(\boldsymbol{a} \mid \boldsymbol{s}) = \mathbb{1}_{\boldsymbol{a}^\star}(\boldsymbol{a}), \quad \text{where } \boldsymbol{a}^\star = \arg\max_{\boldsymbol{a}'} Q(\boldsymbol{s}, \boldsymbol{a}') + \left(\varphi \, \lambda^{1 + \frac{k}{K-1}\alpha}\right) \hat{\sigma}_{\text{epi}}(\boldsymbol{s}, \boldsymbol{a}'). \quad (5)$$

$\lambda \in (0, 1)$ and $\alpha \in \mathbb{R}_{>0}$ are hyperparameters that control the shape of the coefficient distribution. We will refer to this technique as *temporally equalized exploration* (TEE). TEE is inspired by Horgan et al. [44] which uses different values of $\varepsilon$ for the $\varepsilon$-greedy exploration for each actor. In practice, the performances are not sensitive to $\lambda$ and $\alpha$ (see Figure 18 in Appendix G). Both learning and experience collection take place on a single machine and no prioritized experience replay [96] is used since prior work found it ineffective in CMDPs [26].

To summarize, our agent explores the environment in order to gather information about states with high epistemic uncertainty which is measured using ensembles and distributional RL. We build on the algorithm proposed in Clements et al. [20] for estimating the epistemic uncertainty, but we use deep ensemble instead of MAP [87], and use either UCB or Thompson sampling depending on the task and setting[6]. In addition, for UCB, we propose that each actor uses a different exploration coefficient for more diverse data. While variations of the components of our algorithm have been used in prior works, this particular combination is new (see Appendix E). Our ablation experiments show that each design choice is important and **applying these techniques individually or naïvely combining them performs significantly worse.**

## 5 Experiments

### 5.1 Procgen

We compare our method with 3 representative baselines on the standard Procgen benchmark (*i.e.,* 25M steps and easy mode) as suggested by Cobbe et al. [21]: (1) `QR-DQN` which is the prior state-of-the-art value-based method on Procgen [26]; (2) `PPO` which is a popular policy optimization baseline on which most competitive methods are built; and (3) `IDAAC` which is state-of-the-art on Procgen and is built on `PPO`. We tune all hyperparameters of our method on the game `bigfish` only and evaluate all algorithms using the 4 metrics proposed in Agarwal et al. [2]. We run each algorithm on every game for 5 seeds and report the aggregated min-max normalized scores on the full test distribution, and the estimated bootstrap-estimated 95% confidence interval in Figure 4 (simulated with the runs). Our approach significantly outperforms the other baselines in terms of median and interquartile mean (IQM) (which are the more statistically robust metrics according to Agarwal et al. [2]). In particular, it achieves almost 3 times the median score of `QR-DQN` and more than 2 times the IQM of `QR-DQN`. In terms of mean and optimality gap, our method is competitive with `IDAAC` and outperforms all the other baselines. To the best of our knowledge, this is the first value-based method that achieves such strong performance on Procgen. See Appendices I, F, and J for more details about the experiments, the performance on individual games, hyperparameters, and sensitivity analysis.

**Ablations and Exploration Baselines.** We aim to better understand how each component of our algorithm contributes to the final performance by running a number of ablations. In addition, we

---

[6]UCB and Thompson sampling have similar regret guarantees but their actual performance can be problem-dependent [16].

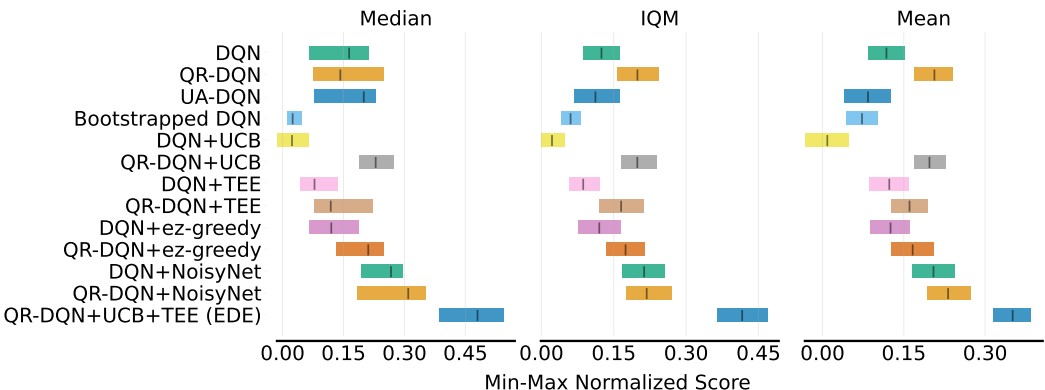

Figure 5: Test performance of different exploration methods on the Procgen benchmark across 5 runs.

compare with other popular exploration techniques for value-based algorithms. Since many of the existing approaches are designed for DQN, we also adapt a subset of them to QR-DQN for a complete comparison. The points of comparison we use are: (1) Bootstrapped DQN [79] which uses bootstrapping to train several copies of models that have distinct exploration behaviors, (2) UCB [19] which uses an ensemble of models to do uncertainty estimation, (3) $\epsilon z$-greedy exploration [24] which repeats the same random action (following a zeta distribution) to achieve temporally extended exploration, (4) UA-DQN [20], and (5) NoisyNet [32] which adds trainable noise to the linear layers [7]. When UCB is combined with QR-DQN, we use the epistemic uncertainty unless specified otherwise. The results are shown in Figure 5. Details can be found in Appendix I.

First, note that both using the epistemic uncertainty via UCB and training on diverse data from TEE are crucial for the strong performance of EDE, with QR-DQN+TEE being worse than QR-DQN+UCB which is itself worse than EDE. Without epistemic uncertainty, the agent cannot do very well even if it trains on diverse data, *i.e.,* EDE is better than QR-DQN+TEE. Similarly, even if the agent uses epistemic uncertainty, it can still further improve its performance by training on diverse data, *i.e.,* EDE is better than QR-DQN+UCB.

Both Bootstrapped DQN and DQN+UCB, which minimize the total uncertainty rather than only the epistemic one, perform worse than DQN, although both are competitive exploration methods on Atari. This highlights the importance of using distributional RL in CMDPs in order to disentangle the epistemic and aleatoric uncertainty. QR-DQN+UCB on the other hand outperforms QR-DQN and DQN because it *only* targets the (biased) epistemic uncertainty. In Figure 16c from Appendix G, we show that indeed exploring with the total uncertainty $\sigma^2$ performs significantly worse at test time than exploring with only the epistemic uncertainty $\sigma^2_{\text{epi}}$. UA-DQN also performs worse than QR-DQN+UCB suggesting that deep ensemble may have better uncertainty estimation (Appendix H).

QR-DQN with $\epsilon z$-greedy exploration marginally improves upon the base QR-DQN, but remains significantly worse than EDE. This may be due to the fact that $\epsilon z$-greedy exploration can induce temporally extended exploration but it is not aware of the agent's uncertainty. Not accounting for uncertainty can be detrimental since CMDPs can have a much larger number of effective states than singleton MDPs. If the models have gathered enough information about a state, further exploring that state can hurt sample efficiency, regardless of whether the exploration is temporally extended.

NoisyNet performs better than the other points of comparison we consider but it is still worse than EDE. The exploration behaviors of NoisyNet are naturally adaptive — the agents will take into account what they have already learned. While a direct theoretical comparison between NoisyNet and our method is hard to establish, we believe adaptivity is a common thread for methods that perform well on CMDPs. Nonetheless, if we consider IQM, none of these methods significantly outperforms one another whereas our method achieves a much higher IQM. Note that TEE cannot be easily applied to NoisyNets and $\epsilon z$-greedy which implicitly use an schedule.

---

[7]Taiga et al. [107] found that, on the whole Atari suite, NoisyNet outperforms more specialized exploration strategies such as ICM [86] and RND [14]. This suggests that these exploration methods may have overfitted to sparse reward environments.

## 5.2 Crafter

To test the generality of our method beyond the Procgen benchmark, we conduct experiments on the Crafter environment [37]. Making progress on Crafter requires a wide range of capabilities such as strong generalization, deep exploration, and long-term reasoning. Crafter evaluates agents using a scalar score that summarizes the agent's abilities. Each episode is procedurally generated, so the number of training environments is practically infinite. While Crafter does not test generalization to new environments, it still requires generalization across the training environments in order to efficiently train on all of them. We build our method on top of the Rainbow implementation [43] provided in the open-sourced code of Hafner [37] and *only* modified the exploration.

We use Thompson sampling instead of UCB+TEE since only one environment is used and Thompson sampling outperforms UCB by itself in this environment. As seen in Table 1, our algorithm achieves significantly higher scores compared to all the baselines presented in Hafner [37], including DreamerV2 [38] which is a state-of-the-art model-based RL algorithm. It is also competitive with LSTM-SPCNN [101] which uses a specialized architecture that does not have spatial pooling with two orders of magnitude more parameters and extensive hyperparameter tuning. The significant improvement over Rainbow, which is a competitive value-based approach, suggests that the exploration strategy is crucial for improving performance on such CMDPs.

Table 1: Results on Crafter after 1M steps and over 10 runs.

| Method | Score (%) |
|---|---|
| EDE (ours) | $11.7 \pm 1.0$ |
| Rainbow | $4.3 \pm 0.2$ |
| PPO | $4.6 \pm 0.3$ |
| DreamerV2 | $10.0 \pm 1.2$ |
| LSTM-SPCNN | $12.1 \pm 0.8$ |

## 6 Related Works

**Generalization in RL.** A large body of work has emphasized the challenges of training RL agents that can generalize to new environments and tasks [93, 67, 50, 84, 118, 121, 77, 21, 22, 49, 58, 36, 18, 12, 13, 35, 53, 3, 26, 66]. This form of generalization is different from generalization in singleton MDP which refers to function approximators generalizing to different states within the same MDP. A natural way to alleviate overfitting is to apply widely-used regularization techniques such as implicit regularization [100], dropout [45], batch normalization [29], or data augmentation [116, 61, 60, 92, 112, 115, 40, 41, 54]. Another family of methods aims to learn better state representations via bisimulation metrics [120, 119, 1], information bottlenecks [45, 28], attention mechanisms [15], contrastive learning [72], adversarial learning [95, 34, 89], or decoupling representation learning from decision making [103, 99]. Other approaches use information-theoretic approaches [18, 71], non-stationarity reduction [46, 78], curriculum learning [47, 109, 48, 85], planning [4], forward-backward representations [111], or diverse policies [57]. More similar to our work, Raileanu and Fergus [90] show that the value function can overfit when trained on CMDPs and propose to decouple the policy from the value optimization to train more robust policies. However, this approach cannot be applied to value-based methods since the policy is directly defined by the Q-function. Most of the above works focus on policy optimization methods, and none emphasizes the key role exploration plays in training more general agents. In contrast, our goal is to understand why value-based methods are significantly worse on CMDPs.

**Exploration.** Exploration is a fundamental aspect of RL [55, 33, 108]. Common approaches include $\varepsilon$-greedy [105], count-based exploration [104, 9, 68], curiosity-based exploration [97, 102, 86], or novelty-based methods specifically designed for exploring sparse reward CMDPs [91, 117, 30, 122]. These methods are based on policy optimization and focus on training agents in sparse reward CMDPs. In contrast, we are interested in leveraging exploration as a way of improving the generalization of value-based methods to new MDPs (including dense reward ones).

Some of the most popular methods for improving exploration in value-based algorithms use noise [82, 32], bootstrapping [79, 81], ensembles [19, 61, 64], uncertainty estimation [80, 83, 20], or distributional RL [70]. This class of method implements the principle of "*optimism in the face of uncertainty*". EDE also falls under this broad class of algorithms. The main goal of these works is to balance return maximization (*i.e.,* exploitation) and exploration to improve sample efficiency on singleton MDPs that require temporally extended exploration. For generalization, both over-exploration and over-exploitation would result in poor generalization in addition to poor training performance, so methods that balance exploration and exploitation would likely be preferred.

Another related area is task-agnostic RL [88, 123] where the agent explores the environment without reward and tries to learn a down-stream task with reward, but to our knowledge, these methods have not been successfully adapted to standard benchmarks like Procgen. Our work is the first one to highlight the role of exploration for faster training on contextual MDPs and better generalization to unseen MDPs. Our work also builds on the distributional RL perspective [10], which is useful in CMDPs for avoiding value overfitting [90].

## 7    Conclusion

In this work, we study how exploration affects an agent's ability to generalize to new environments. Our tabular experiments indicate that effective exploration of the training environments is crucial for generalization to new environments. In CMDPs, exploring an environment is not only useful for finding the optimal policy in that environment but also for acquiring knowledge that can be useful in other environments the agent may encounter at test time. To this end, we propose to encourage exploration of states with high epistemic uncertainty and employ deep ensembles and distributional RL to disentangle the agent's epistemic and aleatoric uncertainties. This results in the first value-based based method to achieve state-of-the-art performance on both Procgen and Crafter, two benchmarks for generalization in RL with high dimensional observations. Our results suggest that exploration is important for all RL algorithms trained and tested on CMDPs. While here we focus on value-based methods, similar ideas could be applied to policy optimization to further improve their generalization abilities. In a broader context, it is perhaps important to emphasize the exploration is not the only piece of puzzle for generalization in RL. There are still environments in Procgen where EDE does not significantly improve the performance even compared to QR-DQN (Figure 13), which indicates that there are other bottlenecks beyond poor exploration. Another limitation of our approach is that it is more computationally expensive due to the ensemble (Appendix H). Thus, we expect it could benefit from future advances in more efficient ways of accurately estimating uncertainty in neural networks.

## Acknowledgement

We would like to thank Alessandro Lazaric, Matteo Pirotta, Samuel Sokota, and Yann Ollivier for the helpful discussion during the early phase of this project. We would also like to thank the members of FAIR London for their feedback on this work and the members of Locus Lab for their feedback on the draft.

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

# A  Sample Complexity of Q-Learning

While the statement that "exploration helps generalization" should be fairly intuitive, the theoretical argument is surprisingly more nuanced. Instead of proving new results, we will rely on several existing results in the literature. To understand why exploration could help the performance of the learned policy under different initial state distributions (*i.e.,* generalization to different initial state distributions), we can study the asymptotic sample complexity of Q-learning in the tabular case, which measures how many steps are required to get a good estimate for all $Q(s, a)$. If the amount of experience is less than the sample complexity, the Q-value can be inaccurate, thus leading to poor performance.

In the theoretical analysis of asynchronous Q-learning, it is common to assume access to a fixed *behavioral policy*, $\pi_b$, which collects data by interacting with the environment and that the environment is ergodic (see all assumptions in Li et al. [63]). The update to the value function is done in an off-policy manner with the data generated by $\pi_b$. Intuitively, the behavioral policy is responsible for exploration. We will first define some useful quantities that relate to the inherent properties of $\pi_b$ and the MDP. Recall that $\mu$ is an initial state distribution and let $p^\pi(\tau \mid s)$ be the trajectory distribution induced by $\pi$ starting from $s$, we can define the state marginal distribution at time $t$ given that the initial state is drawn from $\mu$ as $P_t(s; \pi, \mu) = \mathbb{E}_{\tau \sim p^\pi(\cdot|s_0), s_0 \sim \mu}[\mathbb{1}\{s_t = s\}]$, the state-action marginal distribution $P_t(s, a; \pi, \mu) = \pi(a \mid s)P_t(s; \pi, \mu)$, the stationary state occupancy distribution, $d_\pi(s) = \lim_{T \to \infty} \frac{1}{T+1} \sum_{t=0}^{T} P_t(s; \pi, \mu)$ and finally the stationary state-action distribution $d_\pi(s, a) = \pi(a \mid s)d_\pi(s)$ [8]. Using $D_{\mathrm{TV}}$ to denote the total variation distance, we define:

$$\mu_{\min}(\pi) = \min_{(s,a) \in \mathcal{S} \times \mathcal{A}} d_\pi(s, a), \tag{6}$$

$$t_{\mathrm{mix}}(\pi) = \min \left\{ t \; \middle| \; \max_{(s_0, a_0) \in \mathcal{S} \times \mathcal{A}} D_{\mathrm{TV}}\Big(P_t(s, a; \pi, \delta(s_0, a_0)), d_\pi(s, a)\Big) \leq \frac{1}{4} \right\}. \tag{7}$$

Intuitively, $\mu_{\min}(\pi)$ is the occupancy of the bottleneck state that is hardest to reach by $\pi$, and $t_{\mathrm{mix}}$ captures how fast the distribution generated by $\pi$ decorrelates with any initial state distribution. In this setting, Li et al. [63] showed that with $\tilde{O}\left(\frac{1}{\mu_{\min}(\pi_b)(1-\gamma)^5 \epsilon^2} + \frac{t_{\mathrm{mix}}(\pi_b)}{\mu_{\min}(\pi_b)(1-\gamma)}\right)$ steps[9], one can expect the Q-value to be $\epsilon$-close to the optimal Q-value, *i.e.,* $\max_{(s,a) \in \mathcal{S} \times \mathcal{A}} |Q_t(s, a) - Q^\star(s, a)| \leq \epsilon$ with high probability. The second term of the complexity concerns the rate at which the state-action marginal converges to a steady state, and the first term characterizes the difficulty of learning once the empirical distribution has converged. The crucial point of this analysis is that the sample complexity is *inversely* proportional to the visitation probability of the hardest state to reach by $\pi_b$, so a better exploration policy that aims to explore the state space more uniformly (such as UCB) would improve the sample complexity, which in turn leads to better generalization (*i.e.,* lower Q-value error for all state-action pairs).

The argument above shows that good exploration could ensure that the Q-values are accurate for *any* initial state distribution (which guarantees generalization in a uniform convergence sense), but it does not immediately explain why the Q-values are accurate around the training initial state distribution even though the training initial states further away have inaccurate Q-values, *i.e.,* convergence at these states further away is slower. It turns out that this is related to resetting. It is perhaps worth noting that the analysis so far deals with the continuous RL (*i.e.,* not episodic and does not have an end state). To account for the effect of reset distribution, we may assume that after the episode ends (*e.g.,* reaching the goal in Figure 2), the agent is reset to a state drawn from the initial state distribution or that the agent may be reset to a state drawn from the initial state distribution at every timestep. We use this the continuous MDP as a proxy for the original MDP [10]. In this setting, $d_\pi(s)$ and consequently $d_\pi(s, a)$ will naturally have a higher density around the initial state distribution, so the Q-values at states further away from the initial state would converge slower, leading to these states having poor return even though the policy has high return around the states in the training initial state

---

[8]Notice that the stationary state-action distribution is independent of $\mu$ since it is the stationary distribution of a Markov chain; however, it could still be close to a particular $\mu_{\mathrm{reset}}$ if the environment *resets* to $\mu_{\mathrm{reset}}$ and $\pi_b$ does not explore efficiently (*i.e.,* the Markov chain has a high probability of visiting the initial state distribution).

[9]$\tilde{O}$ hides poly-logarithmic factors.

[10]If the original episodic MDP is ergodic excluding the end state, then the continuous MDP will also be ergodic; however, this modified continuous MDP, in general, does not have the same optimal policy as the original episodic MDP, but it may be close for many common MDPs.

distribution. In other words, we can roughly say that the more a state-action pair is visited, the more likely that its Q-value has a low error for many MDPs.

To further formalize this intuition, we will leverage the concept of *sub-MDP* [51], where we prune $(s, a)$ pairs from $\mathcal{M}$ based on the behavioral policy. Kearns and Singh [51] prunes $\mathcal{M}$ based on whether $d_\pi(s, a)$ is larger than $\alpha$, a hyperparameter that is larger than $\mu_{\min}(\pi_b)$, and argues that one can obtain better sample complexity on the sub-MDP. This can be seen as *throwing away* state-action pairs that are not likely visited by the behavioral policy in the analysis. Concretely, the sub-MDP $\mathcal{M}_\pi(\alpha)$ is an MDP where the state space is:

$$\mathcal{G}(\alpha) = \{(s, a) \in \mathcal{S} \times \mathcal{A} \mid d_\pi(s, a) > \alpha\}. \tag{8}$$

If we further assume that all the state-action pairs visited by the optimal policy, $\pi^\star$, starting from $\mu$ are contained within $\mathcal{G}(\alpha)$, then the optimal policy of the $\mathcal{M}_{\pi_b}(\alpha)$ recovers that of the $\mathcal{M}$ starting from $\mu$. This assumption generally holds for many reasonable MDPs (*e.g.,* those with deterministic transition or bounded reward) and exploration policies $\pi_b$ — in the grid world MDP (Figure 2a), the optimal policy starting from the top left only moves right and never visits any other states, and these states are well covered by $\varepsilon$-greedy exploration; for the training initial state, one can disregard all states that are not on the first row and still retain the optimal policy. By definition $\alpha > \mu_{\min}(\pi_b)$ on $\mathcal{M}_{\pi_b}(\alpha)$ and thus Q-values would converge to the optimal value on $\mathcal{M}_{\pi_b}(\alpha)$ faster than the full MDP, $\mathcal{M}$. If $\alpha$ is sufficiently larger than $\mu_{\min}(\pi_b)$, the sample complexity on $\mathcal{M}_{\pi_b}(\alpha)$ could be much better than on $\mathcal{M}$, explaining the policy's superior performance near the initial state when the $Q(s, a)$ further away from the initial states have not converged. Note that $\mathcal{M}_{\pi_b}(\alpha)$ is only a construct for making the theoretical argument. The Q-values of the "pruned" state-action pairs are still updated as usual algorithmically. Even if the assumption is not true, the optimal policy on $\mathcal{M}_{\pi_b}(\alpha)$ may still be good enough. If the optimal policy on $\mathcal{M}_{\pi_b}(\alpha)$ or $\alpha$ has to be small for the assumption to approximately hold, then one would need a better behavioral policy to solve the training environment efficiently, much less having a good generalization performance on the test environment.

Finally, Li et al. [63] showed that with high probability $\min_{s,a} \frac{2}{3}N(s, a)/t \leq \mu_{\min}(\pi_b)$ which means that having a behavior policy that encourages visiting states with low visitation count, $N(s, a)$, should result in increasing $\min_{s,a} \frac{2}{3}N(s, a)/t$ and thus increasing $\mu_{\min}(\pi_b)$. This indicates that UCB-based exploration should have a better sample complexity than $\epsilon$-greedy which does not explicitly target the visitation count and can be in general pretty bad at covering large state space. Indeed, for episodic RL, the state-of-the-art algorithms generally rely on UCB-based exploration [25] to ensure good sample complexity, which is more similar to what we do in the tabular tasks. Nonetheless, we focus on the continuous case because it highlights the importance of exploration via $\mu_{\min}$.

# B  Case Study: Simplified Bigfish

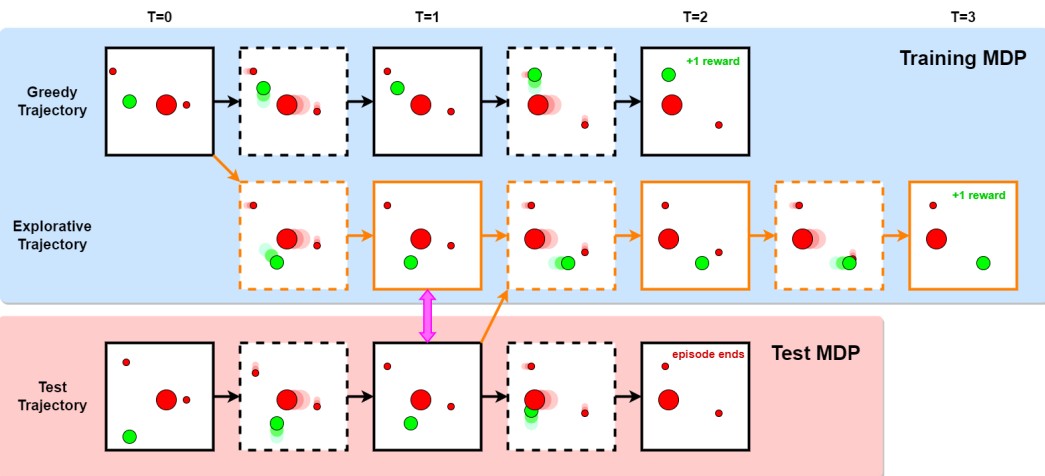

Figure 7: Simplified version of `bigfish`.

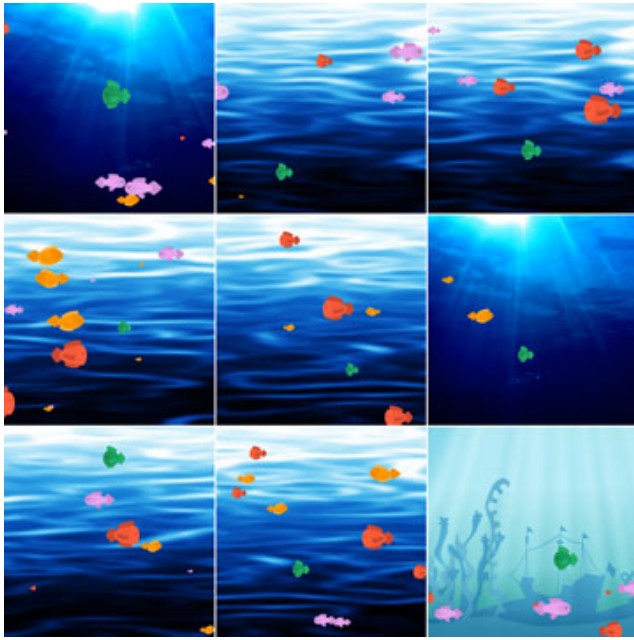

Figure 6: Example frames of `bigfish`. The goal for the agent (green fish) is to avoid fish bigger than itself and eat smaller fish. The spawning of enemy fish is different across different MDPs and the background can change.

To further illustrate the intuition that exploration on the training MDPs can help zero-shot generalization to other MDPs, we will use a slightly simplified example inspired by the game `bigfish` (Figure 6) from Procgen. As illustrated in Figure 7, in `bigfish`, the agent (green circle) needs to eat smaller fish (small red circle) and avoid bigger fish (large red circle). The images with solid borders are the observed frames and the images with dashed borders are the transitions.

If the agent collides with the small fish, it eats the small fish and gains +1 reward; if the agent collides with the big fish, it dies and the episode ends. The blue rectangle (top) represents the training environment and the red rectangle (bottom) represents the test environment. In the training MDP, the agent always starts in the state shown in the $T = 0$ of the greedy trajectory (top row). Using random exploration strategy (*e.g.,* $\epsilon$-greedy), the agent should be able to quickly identify that going to the top will be able to achieve the +1 reward. However, there is an alternative trajectory (middle) where the agent can go down and right to eat the small fish on the right (orange border). This trajectory is longer (therefore harder to achieve via uniformly random exploration than the first one) and has a lower discounted return.

From the perspective of solving the training MDP, the trajectory is suboptimal. If this trajectory is sufficiently hard to sample, the agent with a naive exploration will most likely keep repeating the greedy trajectory. On the other hand, once the agent has sampled the greedy trajectory sufficiently, uncertainty-aware exploration (*e.g.,* UCB or count-based exploration) will more likely sample these rarer trajectories since they have been visited less and thus have higher uncertainty. This has no impact on the performance in the training MDP because the agent has learned the optimal policy regardless of which exploration strategy is used.[11]

However, on the test MDP, the initial state is shown in $T = 0$ of the test trajectory (bottom row). There is no guarantee that the agent knows how to behave correctly in this state because the agent has not seen it during training. One could hope that the neural network has learned a good representation so the agent knows what to do, but this does not necessarily have to be the case — the objective only cares about solving the training MDP and does not explicitly encourage learning representations that help generalization. Suppose that the agent keeps moving up due to randomness in the initialization of the weight, it will run into the big fish and die (bottom row, $T = 2$). Notice that even though the two environments are distinct, the test trajectory (bottom row) and the explorative trajectory (middle row) share the same state at $T = 1$ (pink arrow). Due to this structure, agents that have learned about

---

[11]Using function approximators with insufficient capacity may have some effects on the optimal policy.

the explorative trajectory during training time will be able to recognize that the better action from $T = 1$ is to move to the right which will avoid death and ultimately result in $+1$ reward.

Once again, we emphasize that this is a *simplified* example for illustrative purposes. In reality, the sequences of states do not need to be exactly the same and the neural network can learn to map similar states to similar representations. In principle, the neural network can also learn to avoid the big fish but at $T = 0$ of the training MDP, the behavior "moving up" is indistinguishable from the behavior "avoiding the big fish". Good exploration during training can lead the agent to run into the big fish from all directions which is much closer to "avoid the big fish", the intuitive generalizable behavior. Clearly, not all CMDPs will have this structure which allows us to improve generalization via exploration, but it should not hurt. Like most problems in deep RL, it is difficult to prove this hypothesis analytically but empirically it is indeed the case that EDE as better or the same performance as QR-DQN in 11 out of 16 Procgen environments and perform approximately the same in the remaining environments (Figure 13).

This perspective allows us to tackle the problem with exploration that is in principle orthogonal to representation learning; however, since exploration naturally collects more diverse data, it may also be beneficial for representation learning as a side effect.

## C   Tabular Experiments

### C.1   Generalization to Different Initial States

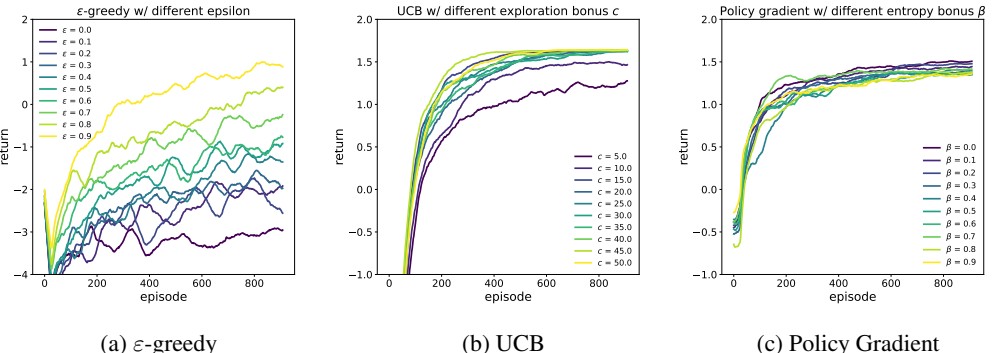

Figure 8: Mean test return for each method with different exploration hyperparameters.

Both the training MDP, $\mu_{\text{train}}$, and the test MDP, $\mu_{\text{test}}$, share the same state space $\mathcal{S}$, $[5] \times [5]$, and action space $\mathcal{A}$, $[4]$, corresponding to {left, right, up, down}. From each state, the transition, $P(s' \mid s, a)$, to the next state corresponding to an action is $100\%$. $\gamma$ is $0.9$. The two MDPs differ by only their initial state distribution: $\rho_{\text{train}}(s) = \mathbb{1}_{(0,0)}(s)$ and $\rho_{\text{test}}(s) = \mathbb{1}_{(4,0)}(s)$.

For both policy gradient and Q-learning, we use a base learning rate of $\alpha_0 = 0.05$ that decays as training progresses. Specifically, at time step $t$, $\alpha_t = \frac{1}{\sqrt{t}}\alpha_0$.

For Q-learning, the Q-function is parameterized as $\vartheta \in \mathbb{R}^{|\mathcal{S}| \times |\mathcal{A}|}$ where each entry is a state-action value. The update is:

$$Q^{(t+1)}(s_t, a_t) = Q^{(t)}(s_t, a_t) + \alpha_t \left( r_t + \gamma \max_a Q^{(t)}(s_{t+1}, a) - Q^{(t)}(s_t, a_t) \right), \qquad (9)$$

where $Q(s, a) = \vartheta_{s,a}$. For Q-learning with UCB, the agent keeps $N(s, a)$ that keeps track of how many times the agent has taken action $a$ from $s$ over the course of training and explores according to:

$$\pi_{\text{ucb}}(a \mid s) = \mathbb{1}_{a^\star}(a) \ \text{ where } \ a^\star = \arg\max_a Q(s, a) + c\sqrt{\frac{\log t}{N(s, a)}}. \qquad (10)$$

For Q-learning with $\varepsilon$-greedy, the exploration policy is:

$$\pi_{\text{egreedy}}(a \mid s) = (1 - \varepsilon)\,\mathbb{1}_{\arg\max_{a'} Q(s, a')}(a) + \varepsilon \left( \sum_{a' \in \mathcal{A}} \frac{1}{|\mathcal{A}|}\mathbb{1}_{a'}(a) \right). \qquad (11)$$

Ties in $\arg\max$ are broken randomly which acts as the source of randomness for UCB.

## C.2 Policy gradient

For policy gradient [114, 106], the policy is parameterized as $\vartheta \in \mathbb{R}^{|\mathcal{S}| \times |\mathcal{A}|}$ where each entry is the *logit* for the distribution of taking action $a$ from $s$, *i.e.,* $\pi(a \mid s) = \frac{\exp(\vartheta_{s,a})}{\sum_{a' \in \mathcal{A}} \exp(\vartheta_{s,a'})}$. The overall gradient is:

$$\nabla_\vartheta \mathbb{E}_{\pi_\vartheta}\left[J(\tau)\right] = \mathbb{E}_{\pi_\vartheta}\left[\sum_{t=0}^{T} \nabla_\vartheta \log \pi_\vartheta(a_t \mid s_t) \sum_{k=t}^{T} \gamma^{k-t} r_k\right], \tag{12}$$

$$\nabla_\vartheta \mathcal{H}_{\pi_\vartheta} = \nabla_\vartheta \mathbb{E}_{\pi_\vartheta}\left[\sum_{t=0}^{T} \sum_{a \in \mathcal{A}} \pi_\vartheta(a \mid s_t) \log \pi_\vartheta(a \mid s_t)\right]. \tag{13}$$

The update is:

$$\vartheta^{(t+1)} = \vartheta^{(t)} + \alpha_t \left(\nabla_\vartheta \mathbb{E}_{\pi_\vartheta}\left[J(\tau)\right] + \beta \nabla_\vartheta \mathcal{H}_{\pi_\vartheta}\right)_{|\vartheta=\vartheta^{(t)}}, \tag{14}$$

where the expectation is estimated with a single trajectory.

For each method, we repeat the experiment for the following 10 hyperparameter values:

- $\varepsilon$: {0.0, 0.1, 0.2, 0.3, 0.4, 0.5, 0.6, 0.7, 0.8, 0.9}
- $c$: {5, 10, 15, 20, 25, 30, 35, 40, 45, 50}
- $\beta$: {0.0, 0.1, 0.2, 0.3, 0.4, 0.5, 0.6, 0.7, 0.8, 0.9}

Each experiment involves training the algorithm for 100 trials. Each trial consists of 1000 episodes and each episode ends when the agent reaches a terminal state or at 250 steps. At test time, all approaches are deterministic: Q-learning follows the action with highest Q-value and policy gradient follows the action with the highest probability. The mean test returns for each experiment are shown in Figure 8. We observe that both UCB and policy gradient are relatively robust to the choice of hyperparameters, whereas $\varepsilon$-greedy's generalization performance varies greatly with $\varepsilon$. Furthermore, even with extremely large $\varepsilon$, the generalization performance is still far worse than UCB and policy gradient. The best-performing values are respectively $\varepsilon^\star = 0.9$, $c^\star = 45$ and $\beta^\star = 0.1$ and the train/test performance of this configuration are shown in Figure 9.PG's performance lies between Greedy and UCB. On average, it converges to a better solution with lower variance than Greedy but it is not as good as UCB.

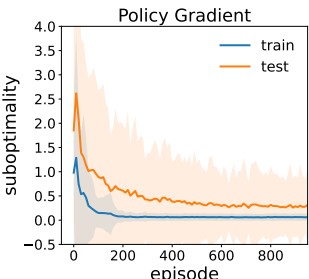

Figure 9: Suboptimality of the best hyperparameters of Policy Gradient

## C.3 Full trajectory update and Sarsa($\lambda$)

Exploration is evidently not the only approach one can take to improve generalization in RL. In fact, we see that policy gradient with rather naive softmax exploration can also outperform $\varepsilon$-greedy exploration (Appendix C.2). We hypothesize that this improvement is due to the fact that these algorithms use whole-trajectory updates (*i.e.,* Monte Carlo methods) rather than one-step updates like Q-learning. To study this, we also consider Sarsa($\lambda$) [105, Ch 12.7] which is an on-policy RL method that learns a Q-value with whole-trajectory updates. The results are shown in Figure 10a.

While both on-policy methods use fairly naive exploration methods (softmax exploration for `PG` and $\varepsilon$-greedy for Sarsa($\lambda$), we observe that both are able to outperform `Greedy`. This observation suggests that their improved performance could be attributed to something else. We believe this is due to the fact that when we use whole-trajectory updates (or n-step returns), the learning signal is propagated to states with low density as soon as a good trajectory is sampled. On the other hand, when we use off-policy approaches, the signal is only propagated when the state action pair is sampled multiple times by the behavioral policy, which may be exponential in the horizon. Nonetheless, there is still a gap between these two methods and `UCB`, suggesting that the generalization benefits of whole-trajectory updates may be distinct from that of exploration. Hence, better exploration can further improve generalization.

### C.3.1 Details of Sarsa($\lambda$)

Sarsa($\lambda$) is an on-policy value-based method that learns Q-value with whole-trajectory update via eligibility traces. Since the algorithmic details are more involved than the other methods we consider, we refer the reader to Chapter 12.7 of Sutton and Barto [105] for the details. The Sarsa($\lambda$) with $\varepsilon$-greedy experiments inherit all the hyperparameters of Q-learning + $\varepsilon$-greedy experiments, but have one more hyperparameter $\lambda \in (0, 1]$ that interpolates the method between Monte Carlo method and one-step TD update. We use a fixed $\lambda = 0.9$ for all experiments. We sweep $\varepsilon$ across 10 values (100 trials each):

- $\varepsilon$:  $\{0.0, 0.1, 0.2, 0.3, 0.4, 0.5, 0.6, 0.7, 0.8, 0.9\}$

It can be seen from Figure 10a that even with whole-episode updates, the average return is still higher for larger exploration coefficients, which means exploration can still help. Furthermore, larger exploration coefficients also tend to have smaller variances in the test performance (Figure 10b). With the best hyperparameter configuration (Figure 10c), Sarsa($\lambda$) outperforms both `Greedy` and `PG`, but is still worse than `UCB` and has a much larger variance.

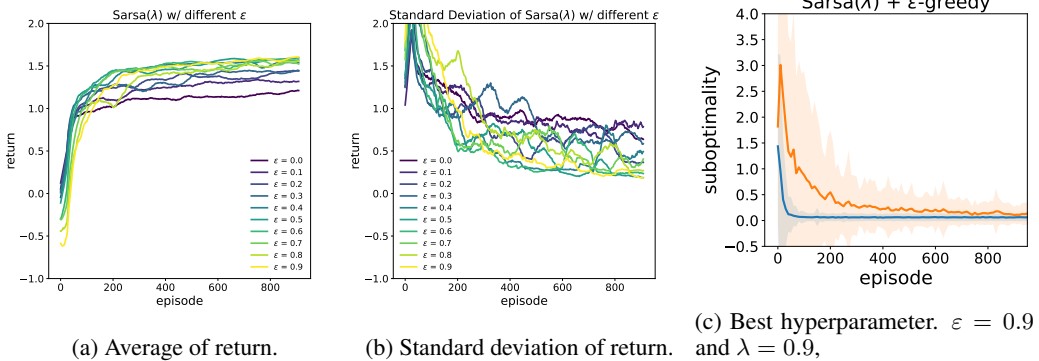

(a) Average of return.    (b) Standard deviation of return.    (c) Best hyperparameter. $\varepsilon = 0.9$ and $\lambda = 0.9$,

Figure 10: Test performance of Sarsa($\lambda$) across different $\varepsilon$. The results of Sarsa($\lambda$). Sarsa($\lambda$) is a value-based method, but it is able to achieve competitive performance because it uses whole-trajectory similar to policy gradient.

### C.4 Generalization to Different Dynamics

In previous section, we presented a simple scenario that exploration helps generalization to different initial state distribution. Here, we demonstrate another scenario where exploration helps generalization to new dynamics. We use a $7 \times 7$ grid as the state space (Figure 11a). The agent receives a small negative reward for every location except for the green square $(4, 2)$, where the agent receives a reward of 2 and the episode ends. The difference between this environment and the previous one is that the agent always starts at $(0, 3)$ but at test time, a gust of wind can blow the agent off course. Specifically, with a probability of $40\%$, the agent will be blown to $+2$ unit in the y-direction. This environment is an example of the *windy world* from Sutton and Barto [105]. The results are shown in Figure 11 and 12, where we see similar results as the previous section, but at test time the optimal expected return is generally worse and has higher variance for all methods since there is stochasticity (suboptimality is computed with respect to the return of an oracle that consistently moves towards

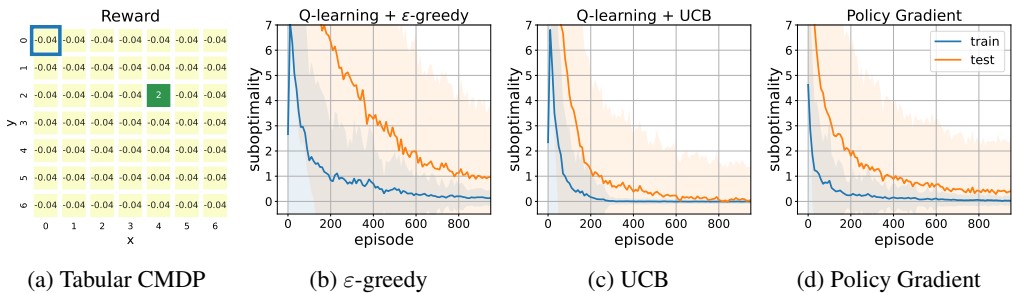

| (a) Tabular CMDP | (b) $\varepsilon$-greedy | (c) UCB | (d) Policy Gradient |

Figure 11: (a) During both training and test time, the agent starts in the blue square, but at test time, with $40\%$ probability, wind can blow the agent down by two unit. In both cases, the goal is to get to the green square. The other plots show the mean and standard deviation of the train and test suboptimality (difference between optimal return and achieved return) over 100 runs for (b) Q-learning with $\varepsilon$-greedy exploration, (c) Q-learning with UCB exploration, and (d) policy gradient with entropy bonus.

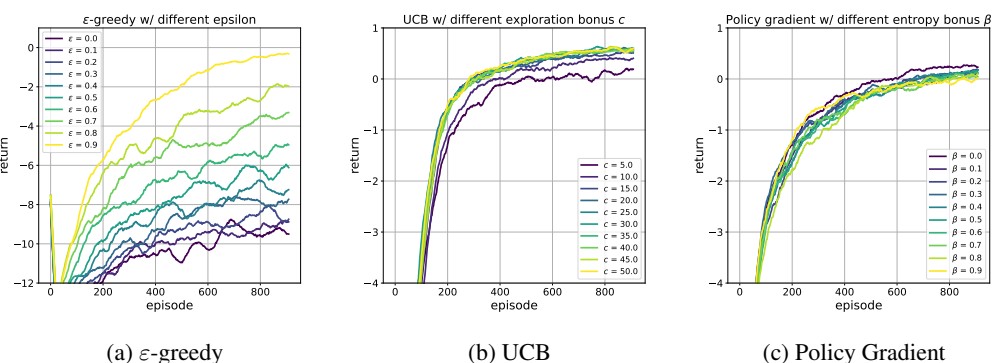

| (a) $\varepsilon$-greedy | (b) UCB | (c) Policy Gradient |

Figure 12: Mean test return for each method with different exploration hyperparameters when the transition dynamic changes at test time.

the goal). This setup is related to *robust reinforcement learning* [76], but exploration makes no assumptions about the type or magnitude of perturbation (thus making more complicated perturbation, *e.g.,* different random initializations of the same game, possible) and does not require access to a simulator.

# D  Additional Background

## D.1  Quantile Regression DQN

Quantile regression DQN (QR-DQN) is a variant of distributional RL that tries to learn the *quantiles* of the state-action value distribution with quantile regression rather than produce the actual distribution of Q-values. Compared to the original C51 [10], QR-DQN has many favorable theoretical properties. Here, we provide the necessary background for understanding QR-DQN and refer the readers to Dabney et al. [23] for a more in-depth discussion of QR-DQN.

Given a random variable $Y$ with associated CDF $F_Y(y) = \mathbb{P}(Y \le y)$, the inverse CDF of $Y$ is

$$F_Y^{-1}(\tau) = \inf \left\{ y \in \mathbb{R} \mid \tau \le F_Y(y) \right\}.$$

To approximate $Y$, we may discretize the CDF with $N$ qunatile values:

$$\left\{ F_Y^{-1}(\tau_1), F_Y^{-1}(\tau_2), \ldots, F_Y^{-1}(\tau_N) \right\} \quad \text{where} \quad \tau_i = \frac{i}{N}.$$

Using the same notation as main text, we use $\boldsymbol{\theta} : \mathcal{S} \times \mathcal{A} \to \mathbb{R}^N$ to denote a parametric function that maps a state-action pair, $(\boldsymbol{s}, \boldsymbol{a})$, to the estimated quantiles $\{\boldsymbol{\theta}_j(\boldsymbol{s}, \boldsymbol{a})\}_{j=1}^N$. The approximated CDF of

$Z(\boldsymbol{s}, \boldsymbol{a})$ is thus:

$$\widehat{Z}(\boldsymbol{s}, \boldsymbol{a}) \stackrel{d}{=} \frac{1}{N} \sum_{j=1}^{N} \delta\left(\boldsymbol{\theta}_j(\boldsymbol{s}, \boldsymbol{a})\right),$$

where $\delta\left(\boldsymbol{\theta}_j(\boldsymbol{s}, \boldsymbol{a})\right)$ is the Dirac delta distribution centered at $\boldsymbol{\theta}_j(\boldsymbol{s}, \boldsymbol{a})$. To learn the quantiles, we use *quantile regression* which allows for unbiased estimate of the true quantile values. Concretely, for a target distribution $Z$ and quantile $\tau$, the value of $F_Z^{-1}(\tau)$ is the minimizer of the quantile regression loss:

$$F_Z^{-1}(\tau) = \arg\min_{\theta} \mathcal{L}_{\mathrm{QR}}^{\tau}(\theta) = \mathbb{E}_{z \sim Z}\left[\rho_\tau(z - \theta)\right] \quad \text{where} \quad \rho_\tau(u) = u\left(\tau - \mathbb{1}(u < 0)\right).$$

Due to discretization, for each quantile $\tau_j$, the minimizer of the 1-Wasserstein distance within that quantile is at $\theta$ that satisfies $F_Z(\theta) = \frac{1}{2}(\tau_{j-1} + \tau_j) = \hat{\tau}_j$ (see Lemma 2 of Dabney et al. [23]). Consequentily, to approximate $Z$, we can simultaneously optimize for all of the quantiles, $\{\theta_j\}_{j=1}^{N}$, by minimizing the following loss:

$$\sum_{j=1}^{N} \mathcal{L}_{\mathrm{QR}}^{\hat{\tau}_j}(\theta_j) = \sum_{j=1}^{N} \mathbb{E}_{z \sim Z}\left[\rho_{\hat{\tau}_j}(z - \theta_j)\right].$$

To stablize optimization, QR-DQN uses an modified version of quantile loss called *quantile Huber loss* which utilizes the Huber loss with hyperparameter $\kappa$:

$$\mathcal{L}_\kappa(u) = \frac{1}{2} u^2 \mathbb{1}(|u| \le \kappa) + \kappa\left(|u| - \frac{1}{2}\kappa\right)\mathbb{1}(|u| > \kappa).$$

The quantile Huber loss is defined as:

$$\rho_\tau^\kappa(u) = |\tau - \mathbb{1}(u < 0)| \mathcal{L}_\kappa(u).$$

In QR-DQN, both the approximating distribution and the target distribution are discretized with quantiles. The target distribution is computed with bootstrapping through the distributional Bellman operator. Concretely, given a target network, $\boldsymbol{\theta}^{\mathrm{target}}$, and transition tuple $(\boldsymbol{s}, \boldsymbol{a}, r, \boldsymbol{s}')$, the target distribution is approximated as:

$$Q(\boldsymbol{s}', \boldsymbol{a}') = \frac{1}{N} \sum_{j=1}^{N} \boldsymbol{\theta}_j^{\mathrm{target}}(\boldsymbol{s}', \boldsymbol{a}')$$

$$\boldsymbol{a}^\star = \arg\max_{\boldsymbol{a}'} Q(\boldsymbol{s}', \boldsymbol{a}')$$

$$\mathcal{T}\boldsymbol{\theta}_j = r + \gamma \boldsymbol{\theta}_j^{\mathrm{target}}(\boldsymbol{s}', \boldsymbol{a}^\star). \tag{15}$$

The TD target Q-value distribution's quantiles and distribution are:

$$\{\mathcal{T}\boldsymbol{\theta}_1, \mathcal{T}\boldsymbol{\theta}_2, \ldots, \mathcal{T}\boldsymbol{\theta}_N\}, \quad \mathcal{T}Z \stackrel{d}{=} \frac{1}{N} \sum_{j=1}^{N} \delta\left(\mathcal{T}\boldsymbol{\theta}_j\right).$$

Treating the target distribution as an oracle (*i.e.,* not differentiable), we update the current quantile estimates to minimize the following quantile regression loss:

$$\sum_{j=1}^{N} \mathbb{E}_{z \sim \mathcal{T}Z}\left[\rho_{\hat{\tau}_j}^\kappa(z - \boldsymbol{\theta}_j(\boldsymbol{s}, \boldsymbol{a}))\right] = \frac{1}{N} \sum_{j=1}^{N} \sum_{i=1}^{N} \rho_{\hat{\tau}_j}^\kappa\left(\mathrm{sg}[\mathcal{T}\boldsymbol{\theta}_i] - \boldsymbol{\theta}_j(\boldsymbol{s}, \boldsymbol{a})\right). \tag{16}$$

sg denotes the stop gradient operation as we do not optimize the target network (standard DQN practice).

# E  Algorithmic Details

In this section, we describe EDE in detail and highlight its differences from previous approaches. In Algorithm 1, we describe the main training loop of EDE. For simplicity, the pseudocode omits the

initial data collection steps during which the agent uses a uniformly random policy to collect data and does not update the parameters. We also write the parallel experience collection as a for loop for easier illustration, but in the actual implementation, all loops over $K$ are parallelized. The main algorithmic loop largely resembles the standard DQN training loop. Most algorithmic differences happen in how exploration is being performed *i.e.,* how the action is chosen and how the ensemble members are updated.

In Algorithm 2, we describe how EDE chooses exploratory actions during training. For each state, we first extract the features with the shared feature extractor $f$ and then compute the state-action value distributions for each ensemble head. With the state-action value distributions, we compute the estimate for the epistemic uncertainty as well as the expected Q-values. Using these quantities, we choose the action based on either Thompson sampling or UCB.

In Algorithm 3, we describe how EDE updates the parameters of the feature extractor and ensemble heads. This is where our algorithm deviates significantly from prior approaches. For each ensemble head, we sample an independent minibatch from the replay buffer and compute the loss like *deep ensemble*. Inside the loop, the loss is computed as if each ensemble member is a QR-DQN. However, doing so naïvely means that the feature extractor $f$ will be updated at a much faster rate. To ensure that all parameters are updated at the same rate, for each update, we randomly choose a single ensemble member that will be responsible for updating the feature extractor and the gradient of other ensemble heads is not propagated to the feature extractor. This also saves some compute. In practice, this turns out to be crucial for performing good uncertainty estimation and generalization. We show the effect of doing so in Figure 17c.

Attentive readers would notice that EDE is more expensive than other methods since it needs to process $M$ minibatches for each update. In our experiments, the speed of EDE with 5 ensemble members is approximately 2.4 times slower than the base QR-DQN (see Appendix H). On the other hand, we empirically observed that the deep ensemble is crucial for performing accurate uncertainty estimation in CMDPs. UA-DQN [20], which uses MAP sampling, performs much worse than EDE even when both use the same number of ensemble members *e.g.,* 3 (see Figure 5). In this work, we do not investigate other alternatives of uncertainty estimation but we believe future works in improving uncertainty estimation in deep learning could significantly reduce the computation overhead of EDE

---

**Algorithm 1** EDE

---

1: Initialize feature extractor $f$ and ensemble heads $\{g_i\}_{i=1}^M$
2: Initialize target feature extractor $f^{\text{target}}$ and ensemble heads $\{g_i^{\text{target}}\}_{i=1}^M$
3: Initialize replay buffer $\texttt{Buffer}$
4: $\{\boldsymbol{s}^{(k)}, \text{done}^{(k)}\}_{k=1}^K \leftarrow$ reset all environment
5: **for** $t$ from 1 to $T$ **do**                       ▷ Standard DQN training loop
6:     **for** $k$ from 1 to $K$ **do**
7:         $\boldsymbol{a} \rightarrow \text{CHOOSEACTION}\big(\boldsymbol{s}^{(k)}, f, \{g_i\}_{i=1}^M, k\big)$
8:         $\boldsymbol{s}', r, \text{done}^{(k)} \leftarrow$ Take $\boldsymbol{a}$ in the environment
9:         add $\big\{\boldsymbol{s}^{(k)}, \boldsymbol{a}, r, \boldsymbol{s}', \text{done}^{(k)}\big\}$ to $\texttt{Buffer}$
10:         $\boldsymbol{s}^{(k)} \leftarrow \boldsymbol{s}'$
11:         **if** $\text{done}^{(k)}$ **then**
12:             $\boldsymbol{s}^{(k)}, \text{done}^{(k)} \leftarrow$ reset the $k^{\text{th}}$ environment
13:         **end if**
14:     **end for**
15:     $\text{UPDATE}\big(\texttt{Buffer}, f, \{g_i\}_{i=1}^M, f^{\text{target}}, \{g_i^{\text{target}}\}_{i=1}^M\big)$
16:     **if** $t \bmod \texttt{update\_frequency} = 0$ **then**
17:         $f^{\text{target}}, \{g_i^{\text{target}}\}_{i=1}^M \leftarrow f, \{g_i\}_{i=1}^M$
18:     **end if**
19: **end for**

---

---

**Algorithm 2** CHOOSEACTION$\big(\boldsymbol{s},\, f,\, \{g_i\}_{i=1}^M, k\big)$

---

1: feature $\leftarrow f(\boldsymbol{s})$            ▷ Only one forward pass on the feature extractor
2: **for** $i$ from 1 to $M$ **do**
3:      $\{\boldsymbol{\theta}_{ij}(\boldsymbol{s}, \boldsymbol{a})\}_{j=1}^N \leftarrow g_i(\texttt{feature})$        ▷ Compute individual ensemble prediction
4: **end for**
5: Compute $\hat{\sigma}_{\text{epi}}^2(\boldsymbol{s}, \boldsymbol{a})$ using Equation 2       ▷ Estimate epistemic uncertainty with ensemble
6: Compute $\boldsymbol{a}^\star$ using Equation 5 or Equation 3       ▷ Explore with estimated uncertainty
7: **return** $\boldsymbol{a}^\star$

---



---

**Algorithm 3** UPDATE$\big(\texttt{Buffer},\, f,\, \{g_i\}_{i=1}^M,\, f^{\text{target}},\, \{g_i^{\text{target}}\}_{i=1}^M\big)$

---

1: $\mathcal{L} \leftarrow 0$
2: $\texttt{grad\_index} \sim \mathcal{U}\{1, M\}$            ▷ Randomly select an ensemble member
3: **for** $i$ from 1 to $M$ **do**          ▷ Train each ensemble member independently
4:      $\{\boldsymbol{s}, \boldsymbol{a}, r, \boldsymbol{s}', \texttt{done}\} \leftarrow$ Sample from $\texttt{Buffer}$
5:      $\texttt{feature}(s') \leftarrow f^{\text{target}}(\boldsymbol{s}')$
6:      $\{\boldsymbol{\theta}_{ij}^{\text{target}}(\boldsymbol{s}', \boldsymbol{a})\}_{j=1}^N \leftarrow g_i^{\text{target}}(\texttt{feature}(s'))$
7:      Compute $\{\mathcal{T}\boldsymbol{\theta}_{i1}, \mathcal{T}\boldsymbol{\theta}_{i2}, \ldots, \mathcal{T}\boldsymbol{\theta}_{iN}\}$ using Equation 15 for the $i^{\text{th}}$ ensemble member
8:      $\texttt{feature}(s) \leftarrow f(\boldsymbol{s})$
9:      **if** $i \neq \texttt{grad\_index}$ **then**          ▷ Prevent over-training the feature extractor
10:          $\texttt{feature}(s) \leftarrow \text{STOPGRADIENT}\,(\texttt{feature}(s))$
11:      **end if**
12:      $\{\boldsymbol{\theta}_{ij}(\boldsymbol{s}, \boldsymbol{a})\}_{j=1}^N \leftarrow g_i(\texttt{feature}(s))$
13:      $\mathcal{L}_i \leftarrow$ compute QR loss using Equation 16 for the $i^{\text{th}}$ ensemble member
14:      $\mathcal{L} \leftarrow \mathcal{L} + \mathcal{L}_i$
15: **end for**
16: Compute gradient of $\mathcal{L}$ w.r.t the parameters of $f$, $\{g_i\}_{i=1}^M$
17: Update the parameters with Adam

---

# F  Procgen Results

In Table 3 and Table 2, we show the mean and standard deviation of the final unnormalized train and test scores (at the end of 25M environment steps) of EDE long side with other methods in the literature. We adapt the tables from Raileanu and Fergus [90]. In Figure 13, we show the curves of min-max normalized test score that we reproduced using the code from Raileanu and Fergus [90]. We see that for a subset of games, EDE s significantly more sample-efficient. For other games, EDE s either on par with the policy optimization methods or fails to train, indicating there are other issues behind its poor performance on these games. Note that IDAAC results tune the hyperparameters for individual games whereas we only tune the hyperparameters on `bigfish`.

| Game | PPO | MixReg | PLR | UCB-DRAC | PPG | DAAC | IDAAC | EDE |
|---|---|---|---|---|---|---|---|---|
| bigfish | $3.7 \pm 1.3$ | $7.1 \pm 1.6$ | $10.9 \pm 2.8$ | $9.2 \pm 2.0$ | $11.2 \pm 1.4$ | $17.8 \pm 1.4$ | $18.5 \pm 1.2$ | $\mathbf{22.1 \pm 2.0}$ |
| StarPilot | $24.9 \pm 1.0$ | $32.4 \pm 1.5$ | $27.9 \pm 4.4$ | $30.0 \pm 1.3$ | $47.2 \pm 1.6$ | $36.4 \pm 2.8$ | $37.0 \pm 2.3$ | $\mathbf{49.6 \pm 2.4}$ |
| FruitBot | $26.2 \pm 1.2$ | $27.3 \pm 0.8$ | $28.0 \pm 1.4$ | $27.6 \pm 0.4$ | $27.8 \pm 0.6$ | $\mathbf{28.6 \pm 0.6}$ | $27.9 \pm 0.5$ | $25.7 \pm 1.4$ |
| BossFight | $7.4 \pm 0.4$ | $8.2 \pm 0.7$ | $8.9 \pm 0.4$ | $7.8 \pm 0.6$ | $\mathbf{10.3 \pm 0.2}$ | $9.6 \pm 0.5$ | $9.8 \pm 0.6$ | $10.0 \pm 0.5$ |
| Ninja | $5.9 \pm 0.2$ | $6.8 \pm 0.5$ | $\mathbf{7.2 \pm 0.4}$ | $6.6 \pm 0.4$ | $6.6 \pm 0.1$ | $6.8 \pm 0.4$ | $6.8 \pm 0.4$ | $6.1 \pm 0.6$ |
| Plunder | $5.2 \pm 0.6$ | $5.9 \pm 0.5$ | $8.7 \pm 2.2$ | $8.3 \pm 1.1$ | $14.3 \pm 2.0$ | $20.7 \pm 3.3$ | $\mathbf{23.3 \pm 1.4}$ | $4.9 \pm 0.5$ |
| CaveFlyer | $5.1 \pm 0.4$ | $6.1 \pm 0.6$ | $6.3 \pm 0.5$ | $5.0 \pm 0.8$ | $7.0 \pm 0.4$ | $4.6 \pm 0.2$ | $5.0 \pm 0.6$ | $\mathbf{7.9 \pm 0.4}$ |
| CoinRun | $8.6 \pm 0.2$ | $8.6 \pm 0.3$ | $8.8 \pm 0.5$ | $8.6 \pm 0.2$ | $8.9 \pm 0.1$ | $9.2 \pm 0.2$ | $\mathbf{9.4 \pm 0.1}$ | $6.7 \pm 0.5$ |
| Jumper | $5.9 \pm 0.2$ | $6.0 \pm 0.3$ | $5.8 \pm 0.5$ | $6.2 \pm 0.3$ | $5.9 \pm 0.1$ | $\mathbf{6.5 \pm 0.4}$ | $6.3 \pm 0.2$ | $5.7 \pm 0.3$ |
| Chaser | $3.5 \pm 0.9$ | $5.8 \pm 1.1$ | $6.9 \pm 1.2$ | $6.3 \pm 0.6$ | $\mathbf{9.8 \pm 0.5}$ | $6.6 \pm 1.2$ | $6.8 \pm 1.0$ | $1.6 \pm 0.1$ |
| Climber | $5.6 \pm 0.5$ | $6.9 \pm 0.7$ | $6.3 \pm 0.8$ | $6.3 \pm 0.6$ | $2.8 \pm 0.4$ | $7.8 \pm 0.2$ | $\mathbf{8.3 \pm 0.4}$ | $5.7 \pm 1.1$ |
| Dodgeball | $1.6 \pm 0.1$ | $1.7 \pm 0.4$ | $1.8 \pm 0.5$ | $4.2 \pm 0.9$ | $2.3 \pm 0.3$ | $3.3 \pm 0.5$ | $3.2 \pm 0.3$ | $\mathbf{13.3 \pm 0.1}$ |
| Heist | $2.5 \pm 0.6$ | $2.6 \pm 0.4$ | $2.9 \pm 0.5$ | $3.5 \pm 0.4$ | $2.8 \pm 0.4$ | $3.3 \pm 0.2$ | $\mathbf{3.5 \pm 0.2}$ | $1.5 \pm 0.3$ |
| Leaper | $4.9 \pm 2.2$ | $5.3 \pm 1.1$ | $6.8 \pm 1.2$ | $4.8 \pm 0.9$ | $\mathbf{8.5 \pm 1.0}$ | $7.3 \pm 1.1$ | $7.7 \pm 1.0$ | $6.4 \pm 0.3$ |
| Maze | $5.5 \pm 0.3$ | $5.2 \pm 0.5$ | $5.5 \pm 0.8$ | $\mathbf{6.3 \pm 0.1}$ | $5.1 \pm 0.3$ | $5.5 \pm 0.2$ | $5.6 \pm 0.3$ | $3.4 \pm 0.5$ |
| Miner | $8.4 \pm 0.7$ | $9.4 \pm 0.4$ | $9.6 \pm 0.6$ | $9.2 \pm 0.6$ | $7.4 \pm 0.2$ | $8.6 \pm 0.9$ | $\mathbf{9.5 \pm 0.4}$ | $0.7 \pm 0.1$ |

Table 2: Procgen scores on test levels after training on 25M environment steps. The mean and standard deviation are computed using 5 runs with different seeds.

| Game | PPO | MixReg | PLR | UCB-DRAC | PPG | DAAC | IDAAC | EDE |
|---|---|---|---|---|---|---|---|---|
| bigfish | $9.2 \pm 2.7$ | $15.0 \pm 1.3$ | $7.8 \pm 1.0$ | $12.8 \pm 1.8$ | $19.9 \pm 1.7$ | $20.1 \pm 1.6$ | $21.8 \pm 1.8$ | $\mathbf{27.5 \pm 2.0}$ |
| StarPilot | $29.0 \pm 1.1$ | $28.7 \pm 1.1$ | $2.6 \pm 0.3$ | $33.1 \pm 1.3$ | $\mathbf{49.6 \pm 2.1}$ | $38.0 \pm 2.6$ | $38.6 \pm 2.2$ | $46.9 \pm 0.7$ |
| FruitBot | $28.8 \pm 0.6$ | $29.9 \pm 0.5$ | $15.9 \pm 1.3$ | $29.3 \pm 0.5$ | $\mathbf{31.1 \pm 0.5}$ | $29.7 \pm 0.4$ | $29.1 \pm 0.7$ | $27.7 \pm 1.0$ |
| BossFight | $8.0 \pm 0.4$ | $7.9 \pm 0.8$ | $8.7 \pm 0.7$ | $8.1 \pm 0.4$ | $\mathbf{11.1 \pm 0.1}$ | $10.0 \pm 0.4$ | $10.4 \pm 0.4$ | $10.6 \pm 0.4$ |
| Ninja | $7.3 \pm 0.3$ | $8.2 \pm 0.4$ | $5.4 \pm 0.5$ | $8.0 \pm 0.4$ | $8.9 \pm 0.2$ | $8.8 \pm 0.2$ | $8.9 \pm 0.3$ | $\mathbf{9.1 \pm 0.4}$ |
| Plunder | $6.1 \pm 0.8$ | $6.2 \pm 0.3$ | $4.1 \pm 1.3$ | $10.2 \pm 1.76$ | $16.4 \pm 1.9$ | $22.5 \pm 2.8$ | $\mathbf{24.6 \pm 1.6}$ | $8.2 \pm 0.8$ |
| CaveFlyer | $6.7 \pm 0.6$ | $6.2 \pm 0.7$ | $6.4 \pm 0.1$ | $5.8 \pm 0.9$ | $9.5 \pm 0.2$ | $5.8 \pm 0.4$ | $6.2 \pm 0.6$ | $\mathbf{10.6 \pm 0.1}$ |
| CoinRun | $9.4 \pm 0.3$ | $9.5 \pm 0.2$ | $5.4 \pm 0.4$ | $9.4 \pm 0.2$ | $\mathbf{9.9 \pm 0.0}$ | $9.8 \pm 0.0$ | $9.8 \pm 0.1$ | $6.6 \pm 0.4$ |
| Jumper | $8.3 \pm 0.2$ | $8.5 \pm 0.4$ | $3.6 \pm 0.5$ | $8.2 \pm 0.1$ | $8.7 \pm 0.1$ | $8.6 \pm 0.3$ | $8.7 \pm 0.2$ | $\mathbf{9.0 \pm 0.4}$ |
| Chaser | $4.1 \pm 0.3$ | $3.4 \pm 0.9$ | $6.3 \pm 0.7$ | $7.0 \pm 0.6$ | $\mathbf{10.7 \pm 0.4}$ | $6.9 \pm 1.2$ | $7.5 \pm 0.8$ | $2.2 \pm 0.1$ |
| Climber | $6.9 \pm 1.0$ | $7.5 \pm 0.8$ | $6.2 \pm 0.8$ | $8.6 \pm 0.6$ | $10.2 \pm 0.2$ | $10.0 \pm 0.3$ | $\mathbf{10.2 \pm 0.7}$ | $10.0 \pm 0.3$ |
| Dodgeball | $5.3 \pm 2.3$ | $9.1 \pm 0.5$ | $2.0 \pm 1.1$ | $7.3 \pm 0.8$ | $5.5 \pm 0.5$ | $5.2 \pm 0.4$ | $4.9 \pm 0.3$ | $\mathbf{15.9 \pm 0.3}$ |
| Heist | $7.1 \pm 0.5$ | $4.4 \pm 0.3$ | $1.2 \pm 0.4$ | $6.2 \pm 0.6$ | $\mathbf{7.4 \pm 0.4}$ | $5.2 \pm 0.7$ | $4.5 \pm 0.3$ | $7.2 \pm 0.1$ |
| Leaper | $5.5 \pm 0.4$ | $3.2 \pm 1.2$ | $6.4 \pm 0.4$ | $5.0 \pm 0.9$ | $\mathbf{9.3 \pm 1.1}$ | $8.0 \pm 1.1$ | $8.3 \pm 0.7$ | $9.0 \pm 0.3$ |
| Maze | $\mathbf{9.1 \pm 0.2}$ | $8.7 \pm 0.7$ | $4.1 \pm 0.5$ | $8.5 \pm 0.3$ | $9.0 \pm 0.2$ | $6.6 \pm 0.4$ | $6.4 \pm 0.5$ | $5.7 \pm 0.9$ |
| Miner | $11.7 \pm 0.5$ | $8.9 \pm 0.9$ | $9.7 \pm 0.4$ | $\mathbf{12.0 \pm 0.3}$ | $11.3 \pm 1.0$ | $11.3 \pm 0.9$ | $11.5 \pm 0.5$ | $1.1 \pm 0.2$ |

Table 3: Procgen scores on train levels after training on 25M environment steps. The mean and standard deviation are computed using 5 runs with different seeds.

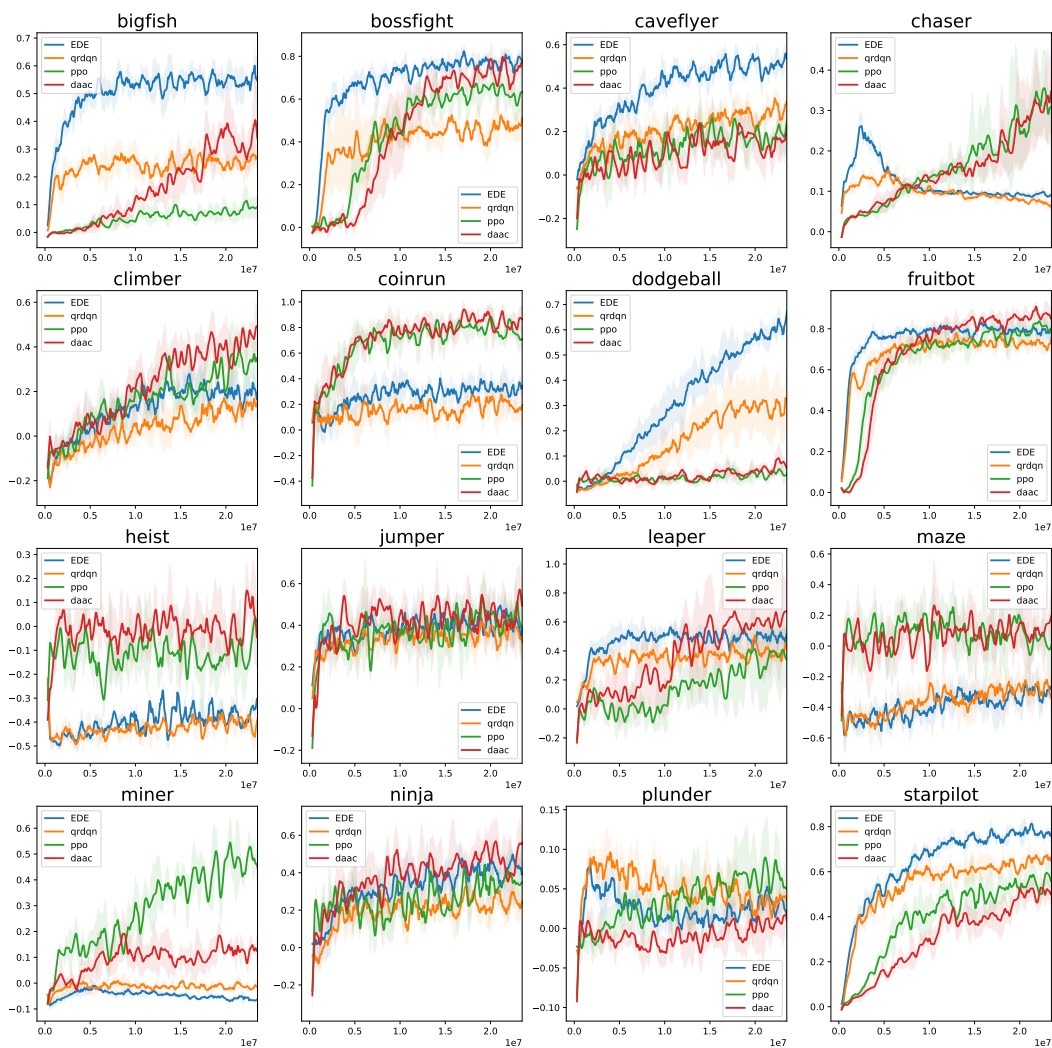

Figure 13: Min-max normalized test performance of a few representative methods on individual Procgen games. Mean and standard deviation are computed over 5 seeds. We can see that for games on which value-based methods can make meaningful progress, EDE almost improve upon QR-DQN. In many cases, EDE is significantly more sample efficient compared to the other methods. There are notably 4 games (chaser, maze, heist, miner) where value-based methods (including EDE) still perform much worse than policy optimization. These games all require generalization with long-horizon planning which appears to remain challenging for value-based methods. Getting value-based methods to generalize on these games would be an interesting future direction.

## F.1 Additional Ablation

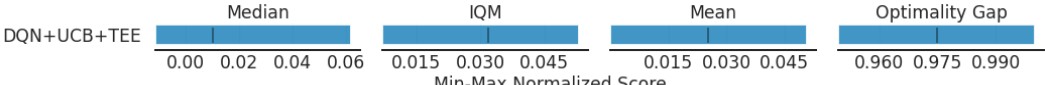

Figure 14: The results on Procgen for DQN+UCB+TEE. The performance is much worse than the other ablations. This is not surprising. Since DQN+UCB performed unfavorably, it is unlikely that TEE would improve its performance.

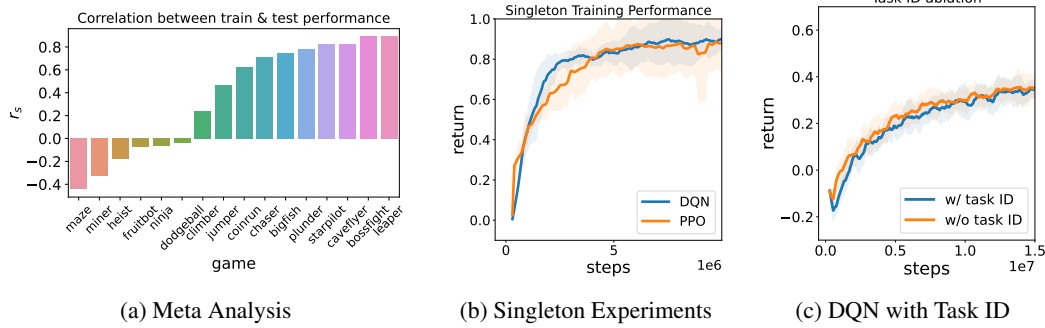

(a) Meta Analysis  (b) Singleton Experiments  (c) DQN with Task ID

Figure 15: Investigations of other potential sources of poor generalization in RL.

## G  Additional Analysis

While our method improves train and test performance on two different CMDP benchmarks, it still does not fully close the generalization gap. In this section, we study other potential reasons for the poor generalization of RL algorithms, particularly value-based ones.

### G.1  Alternative hypothesis

The second hypothesis for the superior performance of UCB relative to Greedy is that better exploration improves sample efficiency resulting in better training performance, which can, in turn, lead to better test performance. Indeed, even in the simple tabular MDPs from Figures 2 and 11, UCB converges faster on the training MDP. Here, we study this hypothesis in more detail. Note that this hypothesis and the one proposed in Section 3 can be *simultaneously true*, but the hypothesis here assumes the model readily learns generalizable representations, whereas the first hypothesis applies even if the agent has perfect state representation (*e.g.,* tabular). In both cases, adapting uncertainty-driven exploration to deep RL on complex CMDPs has additional challenges such as representation learning and unobserved contexts.

**Optimization vs. Generalization.** In supervised learning, deep neural networks have good generalization, meaning that they perform similarly well on both train and test data. However, in RL, the main bottleneck remains optimization rather than generalization (which also boils down to sample complexity). For example in Procgen, the state-of-the-art *training* performance is far below the theoretical optimum [22, 90], suggesting that current RL algorithms do not yet "overfit" the training distribution. To verify this hypothesis, we first conduct a meta-analysis of existing algorithms. We choose 7 representative algorithms (Appendix F) and measure whether better performance on training MDPs translates into better performance on test MDPs. More specifically, we compute the Spearman's rank correlation, $r_s$, between the training and test performance of the algorithms, for each Procgen environment (Figure 15a). For many Procgen environments, training performance is strongly correlated with test performance, suggesting that further improvements in optimization could also result in better generalization.

In Figure 16a and Figure 16b, we show the scatter plot of the training performace and test performance of each algorithm on games with positive correlation and games with negative correlation (each color represents a different game). For games with positive correlation, we see that the correlation is almost linear with a slope of 1 (*i.e.,* almost no generalization gap), which suggests that for these games, we can expect good generalization if the training performance is good. On the games with negative correlation, it is less clear that improving training performance will necessarily lead to better generalization or more severe overfitting. Most of the games with negative correlations are either saturated in terms of performance or too challenging for current methods to make much progress. This suggests that despite their apparent similarity, there are actually two "categories" of games based on the algorithm's tendency to overfit. Note that this dichotomy likely depends on the algorithm; here, all algorithms are PPO-based and we hypothesize the picture would be different for value-based methods. It would be interesting to understand what structural properties of these CMDPs are responsible for this difference in overfitting, but this is out of the scope of the current work.

Another interesting thing to notice is that most of the outliers in the two plots are PLR [48] which is an *adversarial* method that actively tries to reduce training performance. This may explain why the trend does not hold for PLR as all other methods use a uniform distribution over the training environments.

## G.2 Other Sources of Poor Generalization

**Sample efficiency on a single environment.** One hypothesis for the poor performance of value-based methods relative to policy optimization ones in CMDPs is that the former is less sample efficient than the latter in each individual MDP $\mu \in \mathcal{M}_{\text{train}}$. If that is the case, the suboptimality can accumulate across all the training MDPs, resulting in a large gap between these types of methods when trained in all the environments in $\mathcal{M}_{\text{train}}$. To test this hypothesis, we run both DQN and PPO on a *single* environment from each of the Procgen games. As shown in Figure 15b, we see that there is not a significant gap between the average performances of DQN and PPO on single environments (across 3 seeds for 10 environments on which both methods reach scores above 0.5). In fact, DQN is usually slightly more sample-efficient. Thus, we can rule this out as a major factor behind the observed discrepancy.

**Partial observability.** Another sensible hypothesis for why DQN underperforms PPO in CMDPs is the partial observability due to not knowing which environment the agent is interacting with at any given time [35]. Policy optimization methods that use trajectory-wise Monte Carlo update are less susceptible to partial observability whereas value-based methods that use temporal difference updates can suffer more since they rely heavily on the Markovian assumption. To test this hypothesis, we provide the task ID to the Q-network in addition to the observation similar to Liu et al. [65]. The access to the task ID means the agent knows exactly which environment it is in (even though the environment itself may still be partially observable like Atari). In Figure 15b, we see that both methods do well on the singleton environments, so with task IDs, the algorithms should in theory be able to do as well as singleton environments even if there is no transfer between different MDPs because the model can learn them separately. In practice, we embed the discrete task IDs into $\mathbb{R}^{64}$ and add them as input to the final layers of the Q-network. Since there is no semantic relationship between discrete task IDs, we do not expect this to improve generalization performance, but, surprisingly, we found that it *does not* improve training performance either (Figure 15c). This suggests that partial observability may not be the main problem in such environments as far as training is concerned. Note that this issue is related to but not the same as the aforementioned value-overfitting issue. Having task ID means that the agent *can* have an accurate point estimate for each MDP (as far as representation is concerned), but optimization remains challenging without proper exploration.

**Value overfitting.** Most deep RL algorithms model value functions as point estimates of the expected return at each state. As discussed in [90], this is problematic in CMDPs because the same state can have different values in different environments. Hence, the only way for the values to be accurate is to rely on spurious features which are irrelevant for finding the optimal action. For policy optimization algorithms, the policy can be protected from this type of overfitting by using separate networks to train the policy and value, as proposed in [90]. However, this approach cannot be applied to value-based methods since the policy is directly defined by the Q-function. An alternative solution is *distributional RL* [10] which learns a distribution (rather than a point estimate) over all possible Q-values for a given state. This models the *aleatoric uncertainty* resulting from the unobserved contexts in CMDPs (*i.e.,* not knowing which MDP the agent is interacting with), thus being able to account for one state having different potential values depending on the MDP. Our method uses distributional RL to mitigate the problem of value overfitting. As seen in Figure 5, while learning a value distribution leads to better results than learning only a point estimate (*i.e.,* QR-DQN > DQN), the largest gains are due to using a more sophisticated exploration method (*i.e.,* QR-DQN + UCB + TNN $\mathcal{G}(\alpha)$ DQN).

Our analysis indicates that sample efficiency and partial observability are not the main reasons behind the poor generalization of value-based methods. In contrast, value overfitting is indeed a problem, but it can be alleviated with distributional RL. Nevertheless, effective exploration proves to be a key factor in improving the train and test performance of value-based approaches in CMDPs. Our results also suggest that better optimization on the training environment can lead to better generalization to new environments, making it a promising research direction.

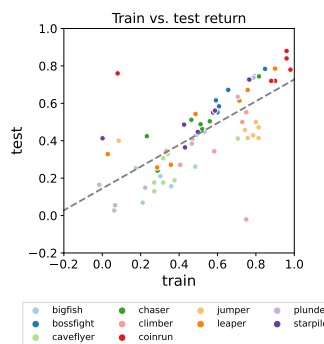

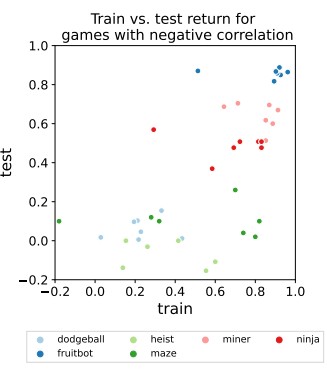

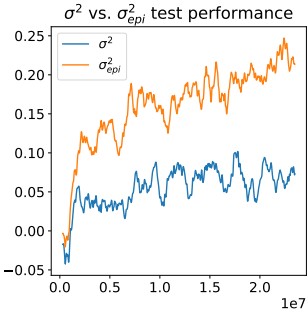

(a) Games with positive train-test correlation.

(b) Games with negative train-test correlation.

(c) Comparing different uncertainties (mean test min-max normalized scores).

Figure 16: Additional empirical results.

# H Uncertainty

Central to our method is the estimation of uncertainty in Q-values. We found that what kind of uncertainty to model and how we model it are crucial to the performance of EDE The first important consideration is what kind of uncertainty to model. Specifically, we argue that only epistemic uncertainty is useful for exploration since they represent the reducible uncertainty. This is a longstanding view shared by many different sub-field of machine learning [74, 98]. The aleatoric uncertainty on the other hand is intrinsic to the environment (*e.g.,* label noise) and cannot be reduced no matter how many samples we gather. In standard regression, even with ensembles, the estimated uncertainty (*i.e.,* variance) contains both epistemic and aleatoric uncertainty. In Atari, the main source of aleatoric uncertainty comes from the sticky action which has a relatively simple structure. However, in CMDPs such as Procgen, the aleatoric uncertainty coming from having different environments is much more complicated and possibly multimodal, making it important to model them explicitly. Empirically, using aleatoric uncertainty to explore is detrimental to the test performance which may partially explain DQN+UCB's extremely poor performance (Figure 5), even compared to DQN+$\varepsilon$-greedy (*i.e.,* uncertainty estimation based on ensemble of non-distributional DQNs contains both types of uncertainties). To further test this hypothesis, we run QR-DQN+UCB with both aleatoric and epistemic uncertainty and a version with only epistemic uncertainty (Figure 16c). The results show that using only epistemic uncertainty performs significantly better than using both uncertainties in terms of both training and test performance.

Some additional important detail about uncertainty estimation with an ensemble are the number of ensembles and how the members of the ensemble are updated. These details have implications for both the computational complexity of inference and the quality of estimated uncertainty. For the main experiments, we use 5 ensemble members which, without specific optimization, results in about 2.5 times more time per algorithm step. The fol lowing are the time complexity of our algorithm on a Tesla V100 (note that this only occurs at the training time since at test time all heads share the same feature extractor):

- EDE 5 ensemble members):     0.3411 seconds per algorithm step

- EDE 3 ensemble members):     0.2331 seconds per algorithm step

- QR-DQN:     0.1356 seconds per algorithm step

In Figure 17b, we found that using 3 ensemble heads achieves comparable performance. On the other hand, using deep ensemble seems to be crucial for a good estimation of epistemic uncertainty as UA-DQN [20], which uses MAP sampling with 3 ensemble members, performs much worse than QR-DQN with 3 ensemble members trained with different minibatch and initialization (which are presumably more independent from each other). We believe further understanding this phenomenon would be an exciting future research direction.

# I   Architecture and Hyperparameters

## I.1   Procgen

**Architecture.**   We use a standard ResNet [42] feature extractor ($f$) from IMPALA [27], which was adopted by Ehrenberg et al. [26] for value-based methods. The feature maps are flattened ($d = 2048$) and processed by $M$ copies of 2-layer MLPs with the same architecture to output $M \times |\mathcal{A}| \times N$ values (for other models, this shape is changed accordingly for the appropriate output dimension). $|\mathcal{A}| = 15$ for all Procgen games. Each MLP conisists of two liner layers (both of dimension 512) separated by a ReLU activation; the first layer is $g^{(1)} : \mathbb{R}^d \to \mathbb{R}^{512}$ and the second layer is $g^{(2)} : \mathbb{R}^{512} \to \mathbb{R}^{|\mathcal{A}| \times N}$. The overall archetecture is

$$g \circ f = g^{(2)} \circ \texttt{ReLU} \circ g^{(1)} \circ \texttt{Flatten} \circ f.$$

The hyperparameters for the main method is shown in Table 4, and are directly adapted from Ehrenberg et al. [26] other than those of new algorithmic components.

For hyperparameters specific to EDE, we search over the following values on the game `bigfish`:

- $\varphi : \{10, 20, 30, 50\}$
- $\lambda : \{0.5, 0.6, 0.7\}$
- $\alpha : \{6, 7, 8, 9\}$

and picked the best combination.

| Name | Value |
|---|---:|
| number of training envs | 200 |
| procgen distribution mode | easy |
| minibatch size | 512 |
| number of parallel workers (K) | 64 |
| replay buffer size | $10^6$ |
| discount factor ($\gamma$) | 0.99 |
| $n$-step return | 3 |
| frame stack | 1 |
| dueling network | no |
| target network update frequency | 32000 (algo step) |
| initial random experience collection steps | $2000 \times K$ (env step) |
| total number of steps | $25 \times 10^6$ (env step) |
| update per algo step | 1 |
| env step per algo step | K |
| Adam learning rate | $2.5 \times 10^{-4}$ |
| Adam epsilon | $1.5 \times 10^{-4}$ |
| prioritized experience replay | no |
| number of quantiles (N) | 200 |
| number of ensemble heads (M) | 5 |
| Huber loss $\kappa$ | 1 |
| gradient clip norm | 10 |
| $\varphi$ (UCB) | 30 |
| $\lambda$ (TEE) | 0.6 |
| $\alpha$ (TEE) | 7 |

Table 4: Hyperparameters for the main Procgen experiments. One algorithm step (algo step) can have multiple environment steps (env step) due to the distributed experience collection.

**Points of comparisons.**   We try to keep the hyperparameters the same as much as possible and only make necessary changes to keep the comparison fair:

- For base DQN and QR-DQN, we use $\varepsilon$-greedy exploration with decaying $\varepsilon$. Specifically, at every algorithmic step, the epsilon is computed with the following equation:

$$\varepsilon(t) = \min\left(0.1 + 0.9\exp\left(-\frac{t - 2000}{8000}\right), 1\right)$$

- For Bootstrapped DQN [79], we use bootstrapping to train the model and uniformly sample a value head for each actor at the beginning of each episode that is used for the entire episode. We use 5 ensemble members just like the other methods.

- For UCB, we use the estimated epistemic uncertainty when combined with QR-DQN and use the standard variance when combined with DQN [19] and use $\varphi = 30$.

- For TEE without UCB, we use $\varepsilon$-greedy exploration *without* decaying $\varepsilon$ for both QR-DQN and DQN, and EDE ith the same $\lambda$ and $\alpha$.

- For $\epsilon z$-greedy, we use the same $\varepsilon$ decay schedule in combination with $n = 10000$ and $\mu = 2.0$ which are the best performing hyperparameter values from Dabney et al. [24]. Each parallel worker has its own $\epsilon z$-greedy exploration process.

- For NoisyNet, we modify the base fully-connected layer with the noisy implementation [32] with $\sigma = 0.5$.

## I.2   Crafter

For Crafter, we use the default architecture and hyperparameters that are used for the Rainbow experiments in Hafner [37] with the following changes:

- Change the distributional RL component from C51 to QR-DQN with appropriate architecture change and use EDE for exploration

- Change the minibatch size from 32 to 64 to speed up training

- Remove prioritized experience replay

Once again, we emphasize that everything else stays the same, so the performance gain can be largely attributed to the proposed method. For EDE since the default configuration from Hafner [37] is single-thread, we use Thompson sampling instead of UCB and do not use TEE.

For $\varphi$, we searched over $\{0.5, 1.0, 2.0\}$ and picked the best value.

| Name | Value |
|---|---|
| minibatch size | 64 |
| number of parallel workers (K) | 1 |
| replay buffer size | $10^6$ |
| discount factor ($\gamma$) | 0.99 |
| $n$-step return | 3 |
| frame stack | 4 |
| dueling network | no |
| target network update frequency | 8000 (algo step) |
| initial random experience collection steps | 20000 (env steps) |
| total number of steps | $10^6$ (env steps) |
| algo step per update | 4 |
| env step per algo step | 1 |
| Adam learning rate | $6.25 \times 10^{-5}$ |
| Adam epsilon | $1.5 \times 10^{-4}$ |
| prioritized experience replay | no |
| number of quantiles (N) | 200 |
| number of ensemble heads (M) | 5 |
| Huber loss $\kappa$ | 1 |
| gradient clip norm | 10 |
| $\varphi$ (Thompson sampling) | 0.5 |

Table 5: Hyperparameters for the final Crafter experiments.

# J  Sensitivity Study

In this section, we study the sensitivity of the test performance (*i.e.,* aggregated min-max normalized scores) to various hyperparameters on a subset of games. First, *without TEE*, we study the sensitivity to different $\varphi$ (UCB exploration coefficients), different $M$ (number of ensemble members), and whether the feature extractor is trained with gradients from each ensemble member.

For $\varphi$ and $M$, we run the algorithm on: `bossfight, ninja, plunder, starpilot, caveflyer, jumper, bigfish, leaper` for 1 seed and report the aggregated min-max normalized scores in Figure 17a and Figure 17b. We observe that the algorithm is slightly sensitive to the choice of $\varphi$, but not sensitive at all to $M$. This is encouraging since a large amount of computational overhead comes from the ensemble. Note that while smaller $\varphi$ performs the best here, we use a larger value ($\varphi = 30$) in combination with TEE.

In Figure 17c, we show the effect of only training the feature extractor using the gradient from one member of the ensemble at every iteration. The results are computed on: `ninja, plunder, jumper, caveflyer, bigfish, leaper, climber` for 1 seed. We observe that always training the feature extractor leads to lower performance, corroborating our intuition that the feature extractor should be trained at the same speed as the individual ensemble members.

In Figure 18, we study the performance under different hyperparameter values of TEE. We use fixed $M = 5$ and $\varphi = 30$ and vary the values of either $\alpha$ or $\lambda$ while holding the other one fixed. We observe no significant difference across these, suggesting that the algorithm is robust to the values of $\alpha$ and $\lambda$.

# K  Hardware

The experiments are conducted on 2080 and V100 and take approximately 250 GPU days.

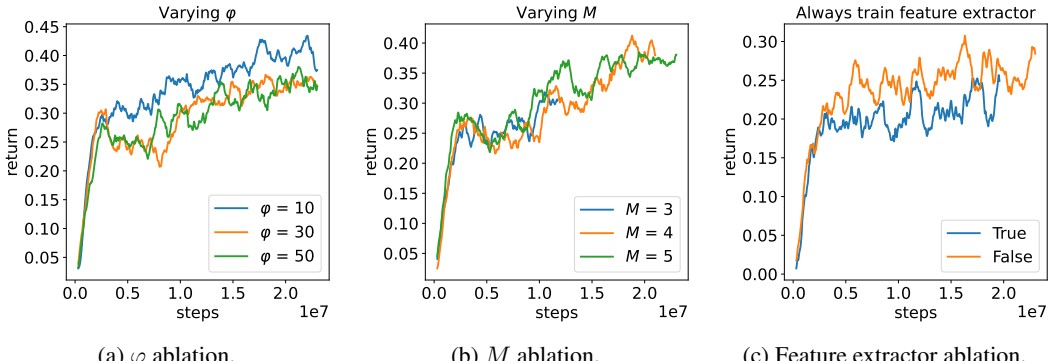

(a) $\varphi$ ablation.

(b) $M$ ablation.

(c) Feature extractor ablation.

Figure 17: Aggregated min-max normalized test scores for $\varphi$ (for fixed $M = 3$, and training feature extractor for all value heads), $M$ (for fixed $\varphi = 50$ and training feature extractor for all value heads), and whether feature extractor is trained with all value head (for fixed $\varphi = 50$ and $M = 3$).

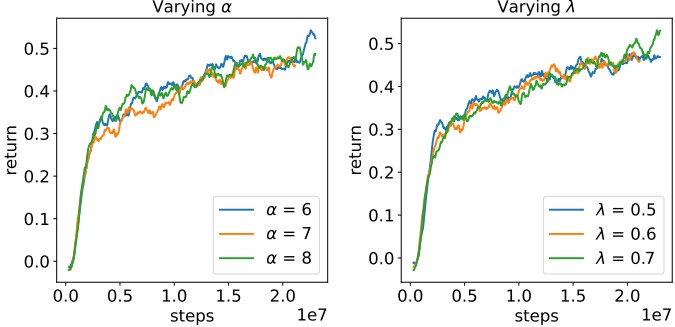

Figure 18: Aggregated min-max normalized test scores for $\lambda$ (for fixed $\alpha = 7$) and $\alpha$ (for fixed $\lambda = 0.6$) on 4 games: `bossfight`, `climber`, `plunder`, `starpilot`.

