# OpenReview forum: "On the Importance of Exploration for Generalization in Reinforcement Learning"
_NeurIPS.cc/2023/Conference — NeurIPS 2023 poster_

### Official Review · Reviewer_dHNU · 2023-06-25

**Soundness:** 3 good
**Presentation:** 4 excellent
**Contribution:** 2 fair
**Rating:** 4
**Confidence:** 5

**Summary:**

The paper motivates the importance of exploration for generalization in contextual MDP with a tabular example, where the context is either the starting state or an uncontrollable random deviations in the transition model. The authors introduce an exploration method for distributional DQN, called EDE, which estimates the epistemic uncertainty of the network as the variance of quantiles between ensemble members. They use this uncertainty in an UCB-style exploration algorithm with varying exploration coefficients, called TEE. EDE/TEE is evaluated on the ProcGen baselines and on the Crafter environment. Results indicate that the average median over test MDP of all environments is higher than baselines with epsilon greedy exploration.


**Strengths:**

The paper is very well written and contains an enormous amount of references, for which the authors are to be commended! The insight that exploration has a larger role for generalization in CMDP than just finding the path to the largest reward (as it is in single MDP) is very interesting and well made. The presented exploration method makes sense and outperforms epsilon-greedy baselines in some of the ProcGen baselines and probably also in Crafter.

**Weaknesses:**

The biggest issue with the paper is that its two parts (exploration for CMDP and EDE) do not really have much to do with each other. EDE/TEE is a general exploration algorithm, which does not use any of the insights from the first part to explore specifically for CMDP. Furthermore, the first part could be more precise for which CMDP exploration is helpful (in the presented analogy), and the second part is not all that novel. Finally, the experiments are missing important baselines and the results of EDE do not seem to be significantly better. To clarify these criticisms:
- The insight given in Figure 1 is very nice, but *only* applies to CMDPs where the context is different initial states. The example given in Figure 11 goes beyond this restriction, but it remains unclear what exactly the class of CMDP that can be solved by exploration is. It appears that either the context is fully observable in the state, in which case the tabular analogy breaks down, or the assumption that all $\mathcal M \sim q_{\mathcal M}$ have the same optimal policy $\pi^*$ does not apply to most CMDP, including the ProcGen benchmarks. It also does not help that the authors phrase uncertainty about an unobserved context as aleatoric uncertainty (which is a reasonable classification), as the existence of $\pi^*$ would make the distinction effectively meaningless.
- EDE is based on the ensemble disagreement (variance) of the output of a distributional RL algorithm (QR-DQN). While the reviewer could not find a paper that does this *exact* combination, it is a very straight forward combination of many papers that pair ensemble-disagreement for exploration (many cited by the authors) with quantile regression DQN. While not in the context of exploration, other papers have already made this combination for risk-avoiding RL (Eriksson et al., 2022; Hoel et al., 2023). In its defense, EDE/TEE uses some non-standard components, like every head being trained by their own minibatch, instead of subsets of the same minibatch as in [78, 80, 81], and different exploration parameters for different actors. These are not new, though, and the overall novelty of EDE/TEE seems low.
- What is the connection between the first and the second part of the paper? It seems that every exploration algorithm would help this type of CMDP generalization, so why did the author specifically propose EDE? This seems to be a missed opportunity, as the first part could be strengthened by a clearer definition of exploration, too. For example, most exploration methods (including TEE) optimize a trade-off between value and epistemic uncertainty, whereas others like task-agnostic exploration [87, 122] only maximize entropy or minimize uncertainty. Which type of exploration is the right one for CMDP generalization?
- Does EDE ever use the aleatoric uncertainty $\sigma_\text{ale}$ in any meaningful way? Is there any significant difference between EDE and a DQN ensemble trained in the same manner? The reviewer doubts that the variance of a DQN ensemble is more tainted by aleatoric uncertainty than one made of QR-DQN. Figure 5 seems to suggest this, but is DQN-UCB/TEE also been trained in the exact same way as EDE?
- It is commendable that the authors ran a range of comparisons in Figure 5, but it is surprising that in Figure 5 EDE seems to have a significant advantage in all quantities, but against PPO/IDAAC there seems to be no advantage in mean and optimality gap. This could be accurate, but the reviewer still recommends the authors to check their code again. Which type of exploration did the PPO and DAAC baselines use here exactly? The reviewer also would have liked a comparison with intrinsic reward methods or UBE [82], as TEE only takes local uncertainty into account, which has proven to be suboptimal for exploration.
- Finally, the conclusions drawn from the presented results appear questionable. Improving over PPO (assumingly without advanced exploration) w.r.t. median but not mean is suspicious and looking at the learning curves in Figure 13 shows that EDE significantly beats PPO in 5 environments, but significantly looses against PPO in another 5 environments (there seems to be no significant difference in the other 6 environments). This does not look like a significant improvement by EDE! Moreover, the story of the paper is that exploration helps generalization. Can you also plot the generalization gaps for all algorithms/environments in Figure 13?

**Additional References**
- H. Eriksson, D. Basu, M. Alibeigi, and C. Dimitrakakis. Sentinel: taming uncertainty with ensemble based distributional reinforcement learning. In Uncertainty in Artificial Intelligence, pages 631-640. PMLR, 2022.
- C.-J. Hoel, K. Wolff, and L. Laine. Ensemble quantile networks: Uncertainty-aware reinforcement learning with applications in autonomous driving. IEEE Transactions on Intelligent Transportation Systems, 2023.

**Questions:**

1. Can you specify precisely under which conditions exploration is useful for tabular CMDP?
2. What is the connection between the first and the second part of the paper? Wouldn't every exploration algorithm help this type of CMDP generalization?
3. Which type of exploration is the right one for CMDP generalization (i.e. for your answer of Q1)?
4. Which type of exploration did the PPO and DAAC baselines use?
5. How does the generalization gaps in Figure 13 look like?


**Limitations:**

It is unclear which exploration is best for CMDPs and whether the performance of EDE is significantly better than the baselines.

---

> ### Author Rebuttal · Authors · 2023-08-09
>
> Thank you for the detailed review and insightful questions! We were glad you found our insight interesting and well made and our literature review extensive.
>
> > Connection between two parts (Q2)
>
> As we explain at the beginning of Sec 4, the first part is meant to show that good exploration can help generalization even in tabular MDPs w/o deep learning. The second part shows that applying this insight to deep RL requires a more careful design of the exploration strategy since empirically not all exploration methods help in more complex environments.
>
> > novelty
>
> We believe that we have been straightforward with the fact that many components of EDE have been studied in prior works. The main claim we are making is that they can be used to improve generalization significantly. We believe this claim is novel and substantiated by our empirical results. Please see the common response for more details on the core contribution of our paper.
>
> > Intrinsic motivation and UBE
>
> Please see our general response. We believe that the baselines are representative of general-purpose exploration methods. For example, UBE belongs to the family of Thomspon sampling algorithms following Bootstrapped DQN.
>
> > Fig 1 and conditions on CMDPs (Q1)
>
> The goal of Fig 1 is a high-level intuition on how exploration can help generalization in RL. Characterizing CMDPs (which is a very large class of problems) is an open area of research. Given the complexity of the problem, we believe it is beyond the scope of our paper. For different initial states, good exploration should generally help. We argue this analytically in Appendix A. For different dynamics, the story is more complicated. “The change in transition is not highly correlated with the action” could be a good condition, but we do not have a precise theoretical characterization. Finally, for different rewards, very strict assumptions are needed to ensure that generalization is possible. Please see [1] for more analysis.
>
> > existence of $\pi^\star$
>
> Thank you for pointing this out. The precise statement [1] is that $\pi^\star$ achieves nearly optimal performance (up to $\alpha$) on all MDP: $V^{s_0}\_{\mathcal{M}}(\pi^\star) \geq \max\_\pi  V\_\mathcal{M}^{s_0}(\pi) - \alpha, \forall \mathcal{M}$ i.e., we assume there exists a policy that does reasonably well on all MDPs of interest without re-training on the unseen MDP. This assumption only ensures that the problem is solvable, but the RL algorithm does not rely on this assumption in any way (i.e., it will find a bad solution if no good solution exists). We will clarify this in the revision.
>
> > Use of aleatoric uncertainty
>
> EDE uses the aleatoric uncertainty by actively ignoring it. This is distinct from risk-sensitive RL which aims to avoid undesirable outcomes by avoiding actions with high aleatoric uncertainty. This is an important topic but not one that we focus on. Future works could combine both (e.g., avoiding high aleatoric uncertainty during test time for better test performance in environments where bad actions can have large penalties).
>
> > EDE vs DQN ensemble
>
> Both DQN ensemble and QRDQN ensemble are not trained on separate minibatches, so DQN+UCB and DQN+TEE are directly comparable to QRDQN+UCB and QRDQN+TEE. We see that QRDQN variants improve upon DQN variants, highlighting the need for aleatoric uncertainty. Without this separation, the variance of DQN ensemble will contain both uncertainties. In Fig 16c, we show an ablation of QRDQN+UCB with both uncertainties and it does significantly worse which means the separation is important.
>
> > Comparison against PPO and checking code
>
> We use the standard rliable [2] for the results so it's unlikely that there is a bug.
>
> > Difference in mean and median
>
> It is well-known in statistics that the median and mean can have different ordering. We have not claimed that EDE outperforms PPO in every game and, the lower mean is because, as you have noticed, there are some games where all value-based (VB) methods perform much worse than policy-based (PB) methods (due to reasons unrelated to exploration) and mean is sensitive to extreme values.
>
> When there are many games, it is customary to report robust summary [2]. Research on Atari shows that rarely does a single algorithm outperform prior works in all games. We agree that it is valuable to understand why VB approaches fail in particualr games, but we don't think this contradicts our claim that the EDE outperforms PPO on Procgen overall, in the sense that the standard RL literature uses the term "*outperform*". We already comment on this in the caption of Fig 14 but would be happy to emphasize it.
>
> > Exploration policy of PPO/DAAC (Q4)
>
> They use the standard Boltzmann policy to explore.
>
> > What kind of exploration (Q3)
>
> The type of exploration we support in this paper is the classical optimism-based that balances value and uncertainty since there is a well-defined task. Please see the common response for more details.
>
> > Generalization gap (Q5)
>
> As pointed out in the footnote of page 3, the notion of generalization we focus on is **test return after a fixed number of interactions** [3] because the generalization gap (gap) can be misleading (e.g., a random policy has 0 gap). In supervised learning, the gap is meaningful because almost all models can fit the training data perfectly, but this is not the case for RL. In terms of gap, EDE is on par with other methods.
>
> **References**
>
> [1] When Is Generalizable Reinforcement Learning Tractable? Malik et al.
>
> [2] Deep Reinforcement Learning at the Edge of the Statistical Precipice. Agarwal et al.
>
> [3] Leveraging Procedural Generation to Benchmark Reinforcement Learning. Cobbe et al.
>
> **Conclusion**
>
> We hope our answers and proposed revision have adequately addressed your questions and concerns and that you will consider raising your score. Please let us know if there is anything preventing you from recommending acceptance and we will respond to any outstanding questions.

---

> > ### Comment · Reviewer_dHNU · 2023-08-10
> >
> > Thanks to the authors for their thorough answers. However, the reviewer still has some doubts and open questions:
> >
> > - Which exploration does QRDQN use in Figure 13?
> >
> > >Our experiments also show that many exploration methods which were designed to improve performance on a single training environment (e.g., NoisyNet, ez-greedy, bootstrapped DQN etc.) are insufficient when applied in such settings. This motivated the design of EDE, [..] we think the algorithmic contribution is secondary to the conceptual message as algorithms can always be improved.
> >
> > >The second part shows that applying this insight to deep RL requires a more careful design of the exploration strategy since empirically not all exploration methods help in more complex environments.
> >
> > > [..] many components of EDE have been studied in prior works. The main claim we are making is that they can be used to improve generalization significantly.
> >
> > - You claim that many exploration methods are insufficient for CMDP generalization. So what *are* the properties of exploration methods that help them to cope with complex CMDPs? Which properties of prior "insufficient" methods are preventing generalization in CMDPs? This is maybe the most important question for the paper, as the authors have made it their main contribution. It is the difference between an interesting observation (workshop contribution) and an insight others can build upon (conference contribution).
> >
> >
> > >We believe that the baselines are representative of general-purpose exploration methods. For example, UBE belongs to the family of Thompson sampling algorithms following Bootstrapped DQN.
> >
> > - I respectfully disagree. UBE accumulates *future uncertainty*, which is different from making decisions as an optimistic trade-off between estimated return and *local uncertainty*, which is what all evaluated methods rely on. UBE is almost identical to intrinsic reward approaches (giving additional reward for local uncertainty) with a separate uncertainty head (which can also use TD($\lambda$) bootstrapping for faster propagation), which are therefore (not only in totally sparse environments) a very different class of exploration methods that should be compared against. Bootstrapped DQN has a similar "long-term" feature to UBE when each ensemble member gets bootstrapped using it's own value prediction (in difference to the average), but operates very differently (choose one value function for the entire episode).
> >
> > >We see that QRDQN variants improve upon DQN variants, highlighting the need for aleatoric uncertainty. In Fig 16c, we show an ablation of QRDQN+UCB with both uncertainties and it does significantly worse which means the separation is important.
> >
> > - QRDQN is known to improve over DQN in many environments, which is consistent over all presented exploration methods in Figure 5  and e.g. in [25]. So far there is no consensus in the community why that is. If I understand Figure 16c correctly, you compare using epistemic uncertainty with epistemic+aleatoric uncertainty as an exploration bonus. Adding additional uncertainty that is (conceptually) not useful for exploration is obviously worse, and not an argument for the claimed distinction between epistemic and aleatoric uncertainty.
> >
> > 	The underlying issue is that, to the detriment of exploration in heteroscedastic environments, epistemic uncertainty estimates *contain* (are "entangled with") aleatoric uncertainty. The variance of an average of $n$ independent random variables $X_i$, with $Var(X_i)=\sigma^2$, is $Var(\frac{1}{n}\sum_{i=1}^n X_i) = \frac{\sigma^2}{n}$. The epistemic (reducible) uncertainty of e.g. averaged returns is therefore proportional to $\frac{1}{n}$, but also contains the aleatoric uncertainty $\sigma^2$. This is a problem if different action, and thus different futures, have different aleatoric uncertainty. See [4] for an (imperfect) attempt to remove aleatoric uncertainty for exploration, that would also make an interesting baseline. Do the authors maintain their claim that ignoring the distributional variance (as an estimate of aleatoric uncertainty) in QRDQN is sufficient to remove the "entanglement" of aleatoric and epistemic uncertainty (which is indeed a problem in exploration)?
> >
> > >Both DQN ensemble and QRDQN ensemble are not trained on separate minibatches
> >
> > - This sounds like a more likely explanation for EDE's performance. Have you run an experiment where e.g. QRDQN+UCB or QRDQN+TEE is trained on separate minibatches? This distinction is important because you claim that EDE, which is mostly a combination of known techniques with some unusual implementation details, is superior. It is therefore important to localize which change improved performance. The reviewer is missing a compelling theoretical justification that explains why EDE explores better, or for which situations it explores better. Can you provide (and substantiate) such an explanation?
> >
> >
> > [4] Nikolov et al. (ICLR 2019). URL https://arxiv.org/abs/1812.07544

---

> > > ### Author Response · Authors · 2023-08-10
> > > **Thank you for the prompt reply! (1/2)**
> > >
> > > Thank you for the prompt and thorough reply, and for engaging with us in the discussion! We are glad that we are able to resolve some of your concerns. We will try to respond to your remaining questions. Once again, we would be happy to discuss further.
> > >
> > > > Which exploration does QRDQN use in Figure 13?
> > >
> > > QRDQN in Figure 13 uses epsilon greedy.
> > >
> > > > So what are the properties of exploration methods that help them to cope with complex CMDPs? Which properties of prior "insufficient" methods are preventing generalization in CMDPs?
> > >
> > > This is a great question. As we pointed out in the paper, we hypothesized that the insufficiency comes from inadequate uncertainty estimation, although we acknowledge that the definition of inadequacy can be nuanced for RL.
> > >
> > > To accomplish this better uncertainty estimation, we first try to separate the two uncertainties. The reasoning is that the source of aleatoric uncertainty in generalization comes from unobserved context which could lead to high variance. We believe this is a rich source of aleatoric uncertainty so modeling them separately is important (perhaps more so than Atari where the main source is sticky action). Of course, as you have noted, distributional RL has other benefits that we do not fully understand, and this separation is not perfect (and hard to achieve), but in practice, it seems to be reasonably effective on the benchmark.
> > >
> > > The second component is using deep ensembles instead of the existing approximate posterior sampling techniques. It is well-known in supervised learning that deep ensembles are much better at uncertainty estimation (e.g., calibration) than traditional techniques because they can better explore different parts of the function space [1]. We hypothesized that this should be beneficial for deep RL too. Note that why deep ensemble works so much better is also largely an empirical observation even though this observation is extremely robust.
> > >
> > >
> > > > UBE accumulates future uncertainty
> > >
> > > As we acknowledged in the general response, it is infeasible for us to compare EDE to all possible baselines in the literature (especially when they have never been applied to Procgen due to the dominance of policy-based methods) and even the same principle can be realized very differently. We are glad that you agree UBE shares important similarities with Bootstrapped DQN (i.e., long-term/deep exploration). We want to note that it's not clear that UBE outperforms bootstrapped DQN even in Atari since the authors explicitly omitted this comparison in their paper. In addition, as far as we are aware, UBE is also not a commonly-used baseline in the exploration literature, whereas bootstrapped DQN is. Bootstrapped DQN is cited 1227 times whereas UBE is cited 181 times and, in our search, we could not find a public implementation of UBE on Atari or a well-known method that uses UBE as their baseline.
> > >
> > > > heteroscedasticity
> > >
> > > If we understand you correctly, you are saying that the estimation of epistemic uncertainty is not accurate due to the sequential nature of MDP, since you agree that using aleatoric uncertainty cannot possibly be helpful for exploration. **At the top of page 6 (starting from line 193), we have already acknowledged that the estimate we are computing is biased due to the nature of RL, but empirically biased estimation can still help in practice if it is informative for exploration. So we have not claimed that ignoring the distributional variance fully resolves the entanglement but only that it alleviates the problem.** We are not the first to make this claim as previous works have observed that this helps for MinAtar [2]. The heteroscedastic noise of RL is indeed a very important problem in RL but it is not the problem that we are trying to solve in this work. We will highlight this more in the revision and add citations accordingly.
> > >
> > > Regarding the effectiveness of the separation (this also relates to your next question), we would like to highlight the change in performance going from QRDQN to QRDQN+UCB and the change in performance going from DQN to DQN+UCB. As you can see, adding UCB to QRDQN improves the median performance (whereas the other two metrics stay relatively the same) but adding UCB to DQN significantly **hurts** the performance. Since they both use the same ensemble technique, we believe that this difference can be attributed to the type of exploration bonus used. In DQN+UCB, we use the variance of different DQN heads which contain both types of uncertainties. In QRDQN+UCB, we use (biased) epistemic uncertainty. While QRDQN generally outperforms DQN, it cannot explain why UCB would lead to worse performance for DQN on Procgen. One sensible hypothesis is that the modeling aleatoric uncertainty is the cause of this difference.

---

> > > > ### Author Response · Authors · 2023-08-10
> > > > **Thank you for the prompt reply! (2/2)**
> > > >
> > > > > This sounds like a more likely explanation for EDE's performance
> > > >
> > > > As we have pointed out above, QRDQN+UCB alone can improve the performance whereas DQN+UCB hurts the performance. This highlights the benefit of modeling biased epistemic uncertainty for exploration in Procgen. Regarding the different minibatch, the goal of doing this is to approximate deep ensembles more closely. Deep ensembles usually train separate networks from different randomness (e.g., initialization and data shuffling), we approximate this with different minibatch for different heads since it’s infeasible to train separate networks for RL. Once again, this is also motivated by better uncertainty estimation via deep ensemble. Bootstrapped DQN already implements something similar where a bootstrapping index is used to make sure the ensemble heads are not trained on the same data from the replay buffer so we don't believe this alone is sufficient given the performance of bootstrapped dqn on procgen.
> > > >
> > > > Due to the computation budget, we were unable to ablate every single component of our algorithm. However, we think our extensive ablation study offers a good understanding of which components matter for Procgen and supports our main claims. We have some preliminary results in Figure 17C that show training the feature extractor with all the minibatches would hurt the performance due to overtraining. In the initial experiments, we found that using separate minibatches is helpful, so we included it in the next iteration of the algorithm. If you feel strongly about this, we can highlight this in the revision and run the full sweep ablation for the final version of the paper.
> > > >
> > > > >  The reviewer is missing a compelling theoretical justification that explains why EDE explores better, or for which situations it explores better
> > > >
> > > > We do not claim that EDE always performs better than other exploration methods and have explained why characterizing which situation it explores better is hard and offered some hypotheses about when it helps in our response. From Figure 13, we can see that EDE improves over QRDQN with epsilon greedy in 11 out of 16 games in Procgen, does the same in 4 out of 16 games, and does slightly worse in 1, so we believe it’s fair to say that the empirical results are fairly robust. This is not a theoretical paper and we have not made any theoretical claim, so we don’t believe it’s reasonable to expect a rigorous regret proof for why EDE does better. To the best of our knowledge, a regret guarantee is not (nor should it be) a necessary condition for accepting deep RL papers to conferences provided that the motivation and empirical results are convincing.
> > > >
> > > > **Reference**
> > > >
> > > > [1] Deep Ensembles: A Loss Landscape Perspective. Fort et al.
> > > >
> > > > [2] Estimating Risk and Uncertainty in Deep Reinforcement Learning. Clements et al.

---

> > > > > ### Author Response · Authors · 2023-08-12
> > > > > **Update on the minibatch**
> > > > >
> > > > > We would like to correct the claim we made about:
> > > > >
> > > > > > Both DQN ensemble and QRDQN ensemble are not trained on separate minibatches
> > > > >
> > > > > We went back to doublecheck the source code and logs, and found that all ensembles are in fact trained with different minibatches. In `level_reply/algo/policy.py` (the source code we provided), you can see in `_loss_qrdqn_bootstrap_multi_head` and `_loss_bootstrap_dqn` that both ensemble has a for loop over `n_ensemble` and draw independent batches from the replay buffer for each member of the ensemble (the `bootstrap` here refer to the fact that they use different minibatches).
> > > > >
> > > > > Sorry for the incorrect statement earlier but these methods are all directly comparable since they all use different minibatches. That is, it is in fact "QRDQN+UCB trained on separate minibatches". We will make this clearer in the paper.

---

> > > > > > ### Comment · Reviewer_dHNU · 2023-08-13
> > > > > >
> > > > > > Thank you for this clarification. While it does not shed much light on why EDE works well, it does lend more credibility to the paper's claim that the combination of QR-DQN+UBE+TEE is somehow the source of the improvement.

---

> > > > > > > ### Author Response · Authors · 2023-08-16
> > > > > > > **Follow-up on the remaining concerns**
> > > > > > >
> > > > > > > We are glad that we have clarified some aspects of EDE. We want to follow up on your remaining concerns before the discussion period ends. We have provided explanations on why EDE works well for our setting above (and in the paper), so we were wondering why these explanations are not sufficient.
> > > > > > >
> > > > > > > - **The benefit of QRDQN** (in addition to its general improvement over DQN) can be seen from the fact that UCB hurts DQN but improves QRDQN. Our explanation for this is that two semantically similar states in CMDP can have very different value functions due to unobserved context [1] so modeling the aleatoric uncertainty allows the model to not memorize these states with patterns that are not useful for generalization (e.g., the background).
> > > > > > > - **The benefit of deep ensemble** is a well-known phenomenon in deep learning as it allows for much better uncertainty quantification for complex data and models compared to alternatives such as Monte-Carlo dropout, variational inference, or local Gaussian approximations because deep ensemble is able to explore more diverse modes in the function space which is beneficial for uncertainty quantification [2]. This is supported by the comparison between UA-DQN which uses local Gaussian approximation and QRDQN+UCB which uses deep ensembles.
> > > > > > > - **The benefit of TEE** is that it ensures that the data seen by the model is more uniform and diverse over time, both of which help optimization under changing data distribution, better uncertainty quantification and less overfitting.
> > > > > > >
> > > > > > > ### Clarification of Paper Structure
> > > > > > >
> > > > > > > More broadly, we want to further clarify the connection between section 3 and section 4 because it seems like it’s a central point of discussion between the reviewers. *We have never intended to indicate that section 4 builds on the intuition of the examples shown in section 3.* EDE uses techniques that are unique to deep learning among other things that cannot occur in a simple deterministic tabular environment. Here is our thought process for arriving at EDE and structuring the paper in this way:
> > > > > > >
> > > > > > > 1. The goal of section 3 is to highlight why exploration may help generalization in RL, which you have agreed is a novel and useful idea.
> > > > > > > 2. Now, analyzing exploration in CMDPs theoretically is very challenging (even exploration in MDP is an area of active research) so the natural (easier) next question is whether exploration helps generalization in more complex problems.
> > > > > > > 3. Do existing exploration methods help generalization significantly in these problems? The answer is negative from our experiments.
> > > > > > > 4. Why do they not help? We make some assumptions and test our hypotheses which culminate in EDE.
> > > > > > >
> > > > > > > Section 3 is crucial for Section 4 because otherwise there is no prior evidence that exploration should be something that would help generalization but it does not provide algorithmic insight for Section 4. If there is any place in the text that has led you to think we intend to say section 4 is built on the intuition from section 3 (other than that exploration can be helpful for generalization), please let us know so we can revise it. Otherwise, if you agree that this is a reasonable way to think about the paper, could you kindly clarify to other reviewers that Section 4 is not meant to be built on intuition from Section 3?
> > > > > > >
> > > > > > > **Reference**
> > > > > > >
> > > > > > > [1] Decoupling Value and Policy for Generalization in Reinforcement Learning. Raileanu et al.
> > > > > > >
> > > > > > > [2] Deep Ensembles: A Loss Landscape Perspective. Fort et al.

---

### Official Review · Reviewer_zePN · 2023-06-28

**Soundness:** 3 good
**Presentation:** 4 excellent
**Contribution:** 3 good
**Rating:** 7
**Confidence:** 3

**Summary:**

This paper proposes an exploration method for value-based RL for contextual MDPs (CMDP) motivated by the idea that good generalization in RL requires attention to RL specific problems such as exploration. The method uses an ensemble of quantized Q-functions to estimate the epistemic uncertainty about the value function. Then uncertainty estimate is used to conduct upper confidence bound (UCB) exploration based on the estimated distribution of the Q-values. Additionally, the algorithm samples more diverse data by using different exploration weights for the different actors. The proposed method achieves better performance than previous value-based methods on procgen, where it is also competitive with policy optimization-based methods. On crafter, the proposed method achieves performance on par with the state of the art method with much smaller network.

## Acknowledgment

I have read the rebuttal and the following discussion and updated my review accordingly. Concerns raised by other reviewers convinced me to lower my rating by one grade.

**Strengths:**

## Originality
This paper presents a well executed study on generalization for value-based RL algorithms in CMDPs. Other studies of generalization in CMDPs have been presented in previous work, but the focus on value-based RL and exploration seems like a relatively unexplored area. The proposed method is a novel combination of existing ideas from previous work. It is well motivated by the presented analysis and achieves state-of-the-art results for value-based RL in challenging CMDPs, so presenting this combination seems like a potentially important contribution to the RL community.

## Quality
The study of exploration as a driver for generalization is of high quality. It starts from illustrative examples and applies that intuition to more complicated cases. The paper hits a good balance in what level of detail to cover the background information. The proposed method is motivated by the analysis and each of the design decisions are carefully analysed in the empirical section. The empirical results are exceptionally thorough and well presented.

## Clarity
The paper is well written throughout. It explains its ideas at a helpful level of detail. The illustrations and figures are clear and well designed. The results for procgen are presented using a style that has been promoted as the best practice in communicating RL results.

## Significance
Value-based RL is an interesting topic and for various tasks, value-based RL algorithms are found to be state-of-the-art. Therefore, contributions that improve the generalization ability of value-based RL algorithms for CMDPs can be significant contributions to the RL community.

**Weaknesses:**

## Discussion on exploration-exploitation tradeoff
- The paper proposes exploration as a critical component for achieving generalization for CMDPs. Choosing a more exploratory policy always introduces an exploration-exploitation tradeoff, yet such tradeoff is not discussed in the paper.
- The illustrative examples assume a distribution shift between training and test time MDPs. If there was no distribution shift, using a more exploratory policy would result in an exploration-exploitation trade-off that needs to be balanced. It would be good to comment on how the presented intuition is still relevant when there is no distribution shift.

## Other weaknesses
- The results for the Crafter are presented with less detail than the Procgen ones. I would have appreciated learning curves and the median/IQM/... plots for Crafter as well.

**Questions:**

- In CMDP settings, where the training MDP distribution is the same as the testing one, would exploration still be the most important driver of generalization performance?

**Limitations:**

The paper discusses relevant limitations to the proposed algorithm.

---

> ### Author Rebuttal · Authors · 2023-08-09
>
> Thank you for the generous review and strong support of our paper! We were excited to hear you found our paper to be an "important contribution to the RL community" and the empirical results to be "exceptionally thorough and well presented".
>
> >exploration-exploitation tradeoff
> >
> Indeed this is a fundamental problem in RL. In our setting, we don’t think the tradeoff is significantly different from other RL settings. Given a fixed time budget, one would need to choose the extent to which you want to explore before settling on a reasonable solution. Both over-exploration and over-exploitation would result in poor generalization in addition to poor training performance. This is why we chose optimism-based exploration which tries to balance between the two automatically. In EDE, this is controlled by the hyperparameter $\varphi$ (w/o TEE) or $\alpha$ and $\lambda$. That being said, in practice, we observed that EDE is not particularly sensitive to these hyperparameters. You can find this ablation in Figures 17 and 18 in the appendix. We will also add more discussion about this to the paper.
>
> > IQM for crafter
> >
> The baseline results are taken from the original Crafter and other papers that use Crafter which do not use IQM so we wouldn't be able to compare to them.
>
> > Distribution of MDPs (Q1)
> >
> Suppose the full distribution contains the two starting states with equal probability and the training MDP is a single sample from the environment starting from the top left corner, the population performance would be just the average of the two curves (blue and orange) and UCB would still perform much better because the training performance (blue curve) is similar for both methods. In this case, there is no distribution shift and the conclusion still stays the same.
>
> **Conclusion**
>
> We thank you again for your feedback and strong support of the paper. We hope our answers have addressed all your questions but let us know if you have any outstanding concerns.

---

> > ### Comment · Reviewer_zePN · 2023-08-12
> >
> > Thanks for the response.
> >
> > I have read the other reviews and the discussion between the authors and the reviewer dHNU. dHNU has successfully drawn my attention to some weaknesses of the paper, which I had overlooked in my review. As dHNU suggests, I may have been swayed by the excellent presentation to ignore some aspects of the scientific content. I still think the paper is worth acceptance to NeurIPS, and am willing to argue for it. At the same time I would not be massively disappointed if it got rejected. I am currently leaning toward lowering my score slightly, but am waiting to see how the discussion continues to unfold.
> >
> > Please see my responses to dHNU on the thread started by the AC.

---

> > > ### Author Response · Authors · 2023-08-12
> > >
> > > Thank you for getting back to us.
> > >
> > > Since the thread you are referring to is not visible to us and the discussion period is not very long, we were wondering if you could let us know your specific concerns so we can address them. Specifically, if you find any of our responses to Reivewer dHNU not convincing, we would be curious to hear why and happy to discuss them further.
> > > Many of the borught-up weaknesses are already discussed in the paper, but if there are places where we could do better, we would be happy to revise the paper.
> > >
> > > Finally, the paper's goal is to use exploration to improve generalization in RL (particularly for value-based methods, where we do improve upon the baseline significantly), not to solve risk-sensitive RL or heteroscedastic noise in RL. We feel that it is unreasonable to criticize a work for problems that the work did not claim to address in the first place.

---

> > > > ### Comment · Reviewer_zePN · 2023-08-13
> > > >
> > > > Apologies, I did not check the visibility of the threads.
> > > >
> > > > I am still positive about the paper and am arguing for its acceptance. I am considering lowering my score slightly because dHNU's review made me realize that the paper does not quite fit the criteria for a "strong accept".
> > > >
> > > > Here's a summary of where I stand after reading the dHNU's review and the following commentary.
> > > >
> > > > * As dHNU points out, the proposed method does not build on the specific intuitions provided in the analysis in section three. However, section three successfully motivates why one might be interested in exploration as a driver of generalization performance in CMDPs. The paper could be stronger if the motivating example and the following empirical study were more closely linked, but I don't see the relative tenuity of the link as a critical flaw. In my opinion, it is better to keep the motivating example simple and illustrative than to make the example harder to understand in order to fit it to the peculiarities of the full empirical setup of deep RL.
> > > > * dHNU does not find the contribution about generalization in CMDP by itself something that would be worth publishing at NeurIPS. I agree that that observation alone is not worthy of publication at a conference level. That being said, I think that section three provides good motivating evidence on why look at exploration as the driver of generalization in CMDPs and the paper crosses the bar for publication with its other contributions.
> > > > * It would strengthen the paper if comparisons to other exploration methods were provided. However, the paper already includes a wide range of well motivated baselines and ablation studies amounting to a significant computational burden. It is hard to justify more experiments as a bar for acceptance in this case.

---

> > > > > ### Author Response · Authors · 2023-08-13
> > > > >
> > > > > Understood, and thank you for your support. We are glad that you find section 3 effective at motivating exploration for generalization as we do. There is so much that we don’t understand about deep learning so it is unrealistic to expect we can replicate everything in a simple tabular setting without function approximation. Below, we want to add a bit more perspective.
> > > > >
> > > > > Indeed there is a gap between the motivating example and Procgen but it is not meant to capture everything. The techniques we used for EDE are specific to deep RL, so it is not possible to have them in a tabular setting – deep ensemble is unique to neural networks, and TEE is used such that the experience is more uniform so the neural network suffers less from forgetting.
> > > > >
> > > > > Even in the tabular setting, we can see in Figure 18 in the Appendix that, for $\epsilon$-greedy, different levels of $\epsilon$ have a significant impact on the test performance in this setting even tho all $\epsilon$ can solve the training MDP very well (the plot in the main text uses $\epsilon=0.9$ since it generalizes the best). In comparison, UCB does well regardless of the level of exploration bonus. Therefore, even in this simple example, *exploration is generally helpful but not all exploration methods are equal* (which is also true in the deep RL setting).
> > > > > This already indicates that UCB-style exploration (more generally exploration that is based on confidence/uncertainty) would be helpful. Of course, one may say that it is obvious since UCB should always be better than epsilon-greedy, but we want to emphasize that their training performances are almost identical, so exploration using uncertainty has a much more significant impact on generalization.
> > > > >
> > > > > Of course, we also agree that this insight alone may not be sufficient because it is unclear whether the real benchmark is similar to the toy example, so we need the empirical result to show the exploration is indeed a real bottleneck for generalization in more complex problems. For those that have worked on applying RL to procedurally generated environments, it should be extremely surprising and interesting that it is possible to have value-based methods with this level of performance since they were usually not even considered as an option for Procgen due to the poor performance.
> > > > >
> > > > > Another interesting thing about the tabular environment we constructed is that it partially sheds light on why policy-based methods do better than value-based methods. Rather than exploration, we believe that it can be attributed to the Monte-Carlo return. We did not delve into details about this in the main text because our focus is exploration and value-based methods but it may also be useful for future work. If you find this interesting, you can find out more about it in Appendix C.3.

---

### Official Review · Reviewer_saZD · 2023-07-05

**Soundness:** 2 fair
**Presentation:** 3 good
**Contribution:** 2 fair
**Rating:** 5
**Confidence:** 4

**Summary:**

This paper introduces a method called EDE (Ensemble Distributional Exploration) that promotes the exploration of states with high epistemic uncertainty through an ensemble of Q-value distributions. The authors evaluate EDE and compare it to several baselines on Procgen and Crafter.

**Strengths:**

**Originality**
The paper's originality lies in its approach to exploration in reinforcement learning. The authors introduce the concept of using ensembles for uncertainty estimation to guide exploration.

**Quality**
See Weaknesses Section.

**Clarity**
The paper is well-written and organized, making it easy for readers to follow the authors' thought process and understand the methodology and results.

**Significance**
By demonstrating that ensembles for uncertainty estimation can effectively guide exploration, the authors have opened up new possibilities for exploration in reinforcement learning. The paper's focus on improving an existing method for better exploration is particularly relevant in today's context, where there is a growing emphasis on improving the efficiency and effectiveness of reinforcement learning.

**Weaknesses:**

One potential weakness is the increased computational cost associated with the method. The use of an ensemble to guide exploration, while useful, also makes the method more expensive to run. This could limit its applicability in scenarios where computational resources are constrained.

In Figure 5, it would be beneficial to add QR-DQN with Thompson Sampling in the ablation.
Furthermore, in Table 1 it would be more informative to evaluate QR-DQN, and EDE (with UCB and TEE instead of Thompson Sampling).

**Questions:**

1. Could you provide some insight into the decision to use Thompson Sampling in the Crafter experiment? Specifically, why was this approach chosen over UCB combined with TEE? Understanding the rationale behind this choice would be beneficial.

**Limitations:**

The authors adequately addressed the limitations

---

> ### Author Rebuttal · Authors · 2023-08-09
>
> Thank you for taking the time to review our paper and for your valuable feedback! We were glad to hear you think our paper "opens up new possibilities" and is "particularly relevant in today's context".
>
> > computational cost
> >
> We agree that this is an important topic. Estimating uncertainty in an efficient manner is an active area of research in machine learning. In supervised learning, methods such as epinet [1] could significantly reduce the computational cost while providing good uncertainty estimates, so we believe it could also be useful in RL. For this work, we chose to use deep ensembles because it’s a commonly used technique that is robust and easy to implement.
>
> Furthermore, the cost of ensembles occurs mostly at training time, and at test time, the additional cost is much smaller due to the shared feature extractor. Since ensembles can achieve much better results, this cost may be justified depending on the application as it only adds a very small additional cost at inference time.
>
> > Thompson sampling vs UCB
> >
> As mentioned in the paper, for Crafter, the environment is not naturally parallelized so we cannot easily use TEE. In this setting without TEE, we observed that TS outperforms UCB. Since our computational budget was limited, we chose to focus on the method that does better in each setting. In the provided source code, both settings can be used for those interested. For procgen, in our preliminary experiment, QR-DQN+TS+TEE did not perform as well as QR-DQN+UCB+TEE. Since evaluating the entire Procgen suite is expensive (~250 GPU days for all the experiments), we did not conduct the full experiment for TS and chose to conduct other ablation and baselines. For bandits, UCB and TS are actually very similar algorithms (in spirit at least), and they have similar regret but their actual performance can be problem-dependent [2] so it's hard to say one will always outperform the other. We will make this more clear in the revision.
>
> **References**
>
> [1] Epistemic Neural Networks. Osband et al.
>
> [2] An Empirical Evaluation of Thompson Sampling. Chapelle et al.
>
>
> **Conclusion**
>
> We hope our answers above have addressed your questions sufficiently well to alleviate your concerns regarding the paper. Please let us know if there is anything standing between us and a strong recommendation of acceptance.

---

> > ### Comment · Reviewer_saZD · 2023-08-19
> >
> > Thank you for the thoughtful rebuttal, which has addressed most of my concerns. I will continue to maintain my current score at this stage.

---

### Official Review · Reviewer_PttU · 2023-07-06

**Soundness:** 3 good
**Presentation:** 2 fair
**Contribution:** 2 fair
**Rating:** 5
**Confidence:** 3

**Summary:**

The paper proposes that effective exploration is important for generalization and shares a value-based method which gets good generalization performance on procgen and outperforms Rainbow on crafter.

**Strengths:**

originality: The idea to leverage improved exploration to improve generalization is novel/seldomly explicitly tackled.
quality: The evaluation is very high-quality.
clarity: The paper is fairly clear. I like starting with a tabular MDP to motivate that exploration improves generalization. The methods section is dense but clear.

**Weaknesses:**

Figure 1 could be improved to show some abstraction for how the type of exploration they propose can improve generalization.

The method is only evaluated on two domains. It could be improved if evaluated on more domains.



**Questions:**

You say this is the first value-based method to get good performance on procgen. The Q-learning baseline from [1] simply adds learning a value-equivalent model as an auxiliary task (but does not use it for planning) and gets pretty good performance. Can you discuss the difference and how your method does better than theres? Is it still true that yours is the first value-based method to get this type of performance on procgen? MuZero also does quite well and is value-based.

The authors compare to LSTM-SPCNN, note that LSTM-SPCNN is a "specialized architecture"  but LSTM-SPCNN is just a visual-transformer like architecture with PPO? So it seems fairly general. Why is the comparison is there?

Of the *many* exploration algorithms that are available to compare against, why did you choose these particular ones?

While I like the idea of improving generalization by improving exploration, do you provide evidence that bad exploration is indeed the reason for poor generalization?

It's hard to judge this paper without answers to the above questions.

Why did you choose crafter over other domains? I could imagine using the mini-grid environments which seem like more relevant environments since you can explicitly setup experiments that test both exploration and generalization.

[1]: Procedural Generalization by Planning with Self-Supervised World Models

**Limitations:**

There is a single line on limitations (expensive due to ensembling) which is insufficient in my opinion.

Will this method improve exploration & generalization in 3D environments like Mine Craft? How about mazes like mini-grid? Why or why not?

Are there different conditions under which you expect the epistemic uncertainty estimate to be more-or-less informative?

---

> ### Author Rebuttal · Authors · 2023-08-09
>
>
> Thank you for taking the time to review our paper and the detailed questions! We were glad to hear you found the idea novel and the evaluation of very high quality. We hope our answers below will address your remaining concerns.
>
> > Figure 1 could be improved to show some abstraction
>
> Figure 1 is meant as an accessible illustration of one possible intuition/interpretation (which other reviewers have found effective), rather than a full description of our method. We opted for simplicity as we were concerned that illustrating uncertainties would result in an overly-complicated diagram which may confuse readers.
>
> > more domains
> >
> Respectfully, we believe this is quite an extensive evaluation of the algorithm. The procgen benchmark already contains **16 distinct games**. Since this is the standard benchmark for studying generalization in RL, we believe this choice is appropriate. The experiments and ablations for these games alone take more than 250 GPU days which are quite extensive in our opinion. On top of that, Crafter, which was designed to resemble Minecraft, adds another domain that poses very different challenges from Procgen. We believe our experiments on both Procgen and Crafter offer sufficient evidence to support our main claims and emphasize the generality of our approach.
>
> > Q1: QL agent from [1]
>
> Thank you for going through the reference and pointing out this one. The QL agent from [1] performs on par with PLR which is worse than IDAAC [2]. In our experiments, our method outperforms IDAAC which suggests that we should outperform [1] as well. Further, we do not claim that exploration is the only way to improve generalization. The main difference is that the QL agent used in [1] uses additional self-supervised learning objectives. This is orthogonal to our contribution, so combining these methods may lead to further improvements. Also note that [1] uses a different setup than we do. Our work adopts the standard setup with 25M training steps on 200 levels from the ``Easy`` mode which allows for direct comparisons to existing methods, whereas [1] uses 30M training steps on 500 levels from the ``Hard`` mode which is not commonly used in the literature.
>
> > Q2: LSTM-SPCNN
>
> LSTM-SPCNN is not a vision transformer (the transformer-based architecture does worse than LSTM-SPCNN). It is a CNN that does not reduce the spatial dimension via striding (hence the ~100x more parameters than QRDQN’s architecture) because striding would lose information about the backpack in the bottom of the frame in crafter which is important for doing well, so LSTM-SPCNN is specialized for crafter.
>
> > Q3: the choice of baselines
>
> Please see our general response at the top.
>
> > Q4 evidence that bad exploration is indeed the reason for poor generalization
>
> In Figure 16c of the appendix, we showed that if one targets both aleatoric and epistemic uncertainty in EDE, the performance is significantly worse, suggesting that the type of exploration indeed affects performance. Nonetheless, there can be many reasons for poor generalization so even a good exploration policy will not always result in better performance if these other components are not appropriate (e.g., poor optimization).
>
> > Q5 Crafter
>
> Crafter is designed to be an extremely challenging environment that requires, among other things, strong generalization and exploration, so we believe it is very relevant for our study. On the other hand, there is no standard benchmark for generalization in mini-grid that everyone uses. In comparison, Crafter is more challenging and has a standardized benchmark to allow for easy comparison and shows how far we can get just by doing better exploration.
>
> > Will this method improve exploration & generalization in 3D environments like Mine Craft? How about mazes like mini-grid? Why or why not?
>
> We believe that it would help in environments like Minecraft since the structure of Crafter is designed to resemble Minecraft (e.g., crafting tree, resource collection and survivial). The 3D part of Minecraft is more challenging for reasons that are less related to exploration (e.g., representation learning). For mazes, it is less clear whether exploration during training is the main bottleneck (it could still be useful during test time). In procgen, there is a maze game. In this setting, we observed that EDE did not improve QR-DQN significantly. We hypothesize that this is due to the fact that mazes require good long-term planning which requires better representation rather than exploration. We commented on this in the caption of Figure 13 but we would be happy to add more discussion.
>
> > Are there different conditions under which you expect the epistemic uncertainty estimate to be more or less informative?
>
> CMDPs are a large and underexplored class of problems so we do not have an exact theoretical characterization right now about when we can expect epistemic uncertainty to help, but it should not hurt in general (it improves over QR-DQN in almost every game). We would be happy to provide some speculations if you think that's helpful.
>
> For different initial states, good exploration should generally help. We argue this analytically in Appendix A. For different dynamics, the story is more complicated. “The change in transition is not highly correlated with the action” could be a good condition, but we do not have a precise theoretical characterization. Finally, for different rewards, very strict assumptions are needed to ensure that generalization is possible. Please see [1] for more analysis.
>
> **References**
>
> [2] Decoupling Value and Policy for Generalization in Reinforcement Learning. Raileanu et al.
>
>
> **Conclusion**
>
> We hope the above clarifications and proposed revision have adequately addressed all your questions regarding the paper. If there are no outstanding concerns, will you consider raising your score to provide strong support for our paper?

---

> > ### Comment · Reviewer_PttU · 2023-08-15
> > **Reviewer Response**
> >
> > Thank you for your response.
> >
> > > Figure 1 is meant as an accessible illustration of one possible intuition/interpretation (which other reviewers have found effective), rather than a full description of our method. We opted for simplicity as we were concerned that illustrating uncertainties would result in an overly-complicated diagram which may confuse readers.
> >
> > Given other reviewers (e.g. dHNU), I still think Figure 1 is sub-optimal for your goal of motivating exploration for Deep RL.
> >
> > > main difference is that the QL agent used in [1] uses additional self-supervised learning objectives. This is orthogonal to our contribution, so combining these methods may lead to further improvements.
> >
> > I agree. I still think the claim "this is the first model-free value-based method to achieve state-of-the-art performance on these benchmarks" is misleading since they were also a model-free value-based meth. I would remove this from the text. Good point that they study the `hard` setting with 300M training steps instead of 25M training steps.
> >
> > > LSTM-SPCNN is specialized for crafter.
> >
> > I disagree that LSTM-SPCNN is specialized to crafter since it can be applied to any other vision-based RL domain. Though I agree that it was designed to target crafter by using patches with attention. Does some part of its design uses crafter-specific details?
> >
> > While I appreciate the complexity and variety of crafter + procgen for your experiments, I think it does make the connection to section 3 even less clear. I think some visualization across any of the Deep RL domains you study which showcased the benefit of exploration for generalization would dramatically improve this paper.
> >
> > I think reviewer dHNU did a good job of pointing out this papers limitations. I keep my current score.

---

> > > ### Author Response · Authors · 2023-08-16
> > > **Clarifying a misunderstanding**
> > >
> > > Thank you for getting back to us. We would like to first clarify a misunderstanding as it seems to be a central discussion among the reviewers. The goal of figure 1 and section 3 was not to motivate EDE but to motivate why exploration might be helpful for generalization in RL. EDE's algorithmic designs require additional motivation which we explained in the paper. We have never intended to suggest that we derived EDE from section 3 and figure 1. Indeed it would be hard if not impossible to visualize what happens in a neural network in a tabular environment or an illustrating figure. If there are places in the text that may suggest this is the case, please let us know so we can change it.
> > >
> > > The flow of logic is that “*Figure 1 and Section 3 demonstrate that there exist situations in RL where better exploration can help generalization because this is not well-known before -> Section 4 shows that we can design an exploration method that drastically improves generalization in deep RL*”, **not** “*The method in Section 4 works because Figure 1 and Section 3 explains everything that happens in deep RL*”. To the best of our knowledge, exploration was not a well-known avenue for improving generalization in RL, so we wanted to have a didactic example that illustrates this point as straightforwardly as possible.
> > >
> > > If you have suggestions about how we could improve figure 1 without increasing the cognitive load of the reader, we would be happy to incorporate the suggestions.
> > >
> > > > I still think the claim "this is the first model-free value-based method to achieve state-of-the-art performance on these benchmarks" is misleading since they were also a model-free value-based meth. I would remove this from the text.
> > >
> > > We will change the text to “competitive” or “strong” to be more appropriate. However, we want to emphasize that the QL agent in [1] was not competitive even at the time when [1] first came out. In their setting, the QL agent has the same performance as PLR that is worse than IDAAC which came out before [1]. If the behavior transfers between the two settings, then QL agent should have worse performance than EDE as EDE performs better than IDAAC.
> > >
> > > > I disagree that LSTM-SPCNN is specialized to crafter since it can be applied to any other vision-based RL domain. Though I agree that it was designed to target crafter by using patches with attention.
> > >
> > > By specialize we mean exactly that it is designed to target Crafter. We will revise it to “designed to target the properties of Crafter” to avoid ambiguity. To be more specific, “SPCNN does not have pooling layers, so the resulting output tensor is of the same height and width as the input image. The flattened tensor that is fed into a linear layer is much larger (64x64x64 instead of 8x8x64 for CNN).” This is quite an unusual architecture design for most applications because it makes the linear layer at least 64 times larger.
> > >
> > > >  I think some visualization across any of the Deep RL domains you study which showcased the benefit of exploration for generalization would dramatically improve this paper.
> > >
> > > In figure 3 of the main text, we have a visualization that shows the benefit of exploration in the game `bigfish` from Procgen. The environment does not support resetting to an arbitrary state due to procedural generation so we are unable to generate the exact frame.

---

### Official Review · Reviewer_tLzS · 2023-07-07

**Soundness:** 3 good
**Presentation:** 3 good
**Contribution:** 3 good
**Rating:** 7
**Confidence:** 3

**Summary:**

The paper focuses on the importance of exploration in the generalizability in contextual MDPs. The proposed method is built on QR-DQN where the epistemic uncertainty can be separated from aleatoric uncertainty via ensemble. The epistemic uncertainty is then used in a UCB manner to promote exploration. The resulting algorithm is tested in two benchmarks and shown significant improvement over previous baselines.

**Strengths:**

- The paper is overall well-organized and easy to follow.
- The proposed method is novel, well-motivated and empirically strong.


**Weaknesses:**

- The motivating environment introduced in Sec 3 doesn't seem exactly appropriate for the following reasons:
    - According to the definition of CMDP, the distribution of MDPs during training and testing should be the same, which is not the case here.
    - Since only the starting distribution is different, the value function is the same in both MDPs. Combined with above mentioned different distribution during training and testing, this example seems too obviously engineered towards promoting visiting more states during training.
    - Claiming UCB is more effective in exploration compared to $\epsilon$-greedy seems dry. It would be helpful to at least compare the training stage state-action visiting frequency for both methods.
- Same as the motivating example, it would be helpful to show either qualitatively or quantitatively the difference in the extent of exploration among different methods.

**Questions:**

The authors emphasize separating epistemic uncertainty from the overall uncertainty, which motivates them to build their framework on top of QR-DQN, from my understanding. This decision makes sense theoretically. However, there is a lack of empirical study to show the actual significance. If this separation is unimportant, this method can potentially be applied to a wider range of algorithms.

Minor typos / mistakes:
- No legend in Figure 2(b) and 2\(c\).

**Limitations:**

The authors adequately addressed the limitations and potential negative societal impact of their work.

---

> ### Author Rebuttal · Authors · 2023-08-09
>
> Thank you for the thoughtful review and support for the paper! We were glad to hear that you found our paper "well-organized and easy to follow" and our proposed method "novel, well-motivated, and empirically strong". We hope our answers below will address your remaining questions.
>
>
> >The motivating environment introduced in Sec 3
> >
> We agree that the claim that exploration can help generalization to new environments (with different initial states or dynamics) may seem obvious *after* we have put them into the context. In fact, we think that it's good that it's so intuitive since the toy example is meant to provide intuition. But of course, the motivating example is not meant to and cannot capture everything that happens in deep RL.
>
> As far as we know, the observation that exploration affects generalization in RL, has not been explicitly discussed in any prior work, so we believe it is still a valuable contribution to explicitly emphasize this, provide intuition using simplified settings and controlled experiments, and validate this hypothesis in more challenging environments like Procgen and Crafter, which our paper does.
>
> > The distribution of MDPs during training and testing should be the same
> >
> Regarding the distribution of MDPs, the distributions can be easily made the same: the full distribution contains the two starting states with equal probability and the training set contains a single sample from the distribution which is the environment starting from the top left corner. In this case, the population performance would be just the average of the two curves (blue and orange) and UCB would still perform much better since the performance on training MDP (blue) is about the same for both methods. We'd be happy to add a discussion on this point in the appendix.
>
>
> >Importance of separation of different uncertainties
> >
> Our paper contains multiple experiments that support the importance of separating the two types of uncertainties. First, in Figure 16c from the appendix, we directly compare aggregated (both aleatoric and epistemic) with only epistemic uncertainties showing that using only the epistemic uncertainty to guide exploration is significantly better. Second, Figure 5 from the main paper shows that the ensemble of DQN (which contains both uncertainties) performs significantly worse.
>
>
> > Extent of exploration
> >
> It is difficult to visualize exploration in a procedurally generated environment because there could be a potentially infinite number of distinct states. Moreover, we probably do not want the policy to visit all of these states since that would be extremely inefficient.

---

### Author Rebuttal · Authors · 2023-08-09

We thank all the reviewers for taking the time to provide valuable feedback on our work. Overall, the reviewers found our paper to be clear and easy to follow, the idea of using exploration to improve generalization to be novel and well-motivated, and the experimental results to be thorough and strong. Below we clarify the main contribution of the paper and motivate our choice of baselines since multiple reviewers had questions about these.


> Core contribution of the paper

We'd like to emphasize that **the main message of this paper is the insight that appropriate exploration can help generalization to new environments in RL**. This message is quite intuitive on the tabular MDP we constructed, but it is only relevant if it helps empirically on more complex problems such as Procgen or Crafter, which we use to validate this hypothesis. Our experiments also show that many exploration methods which were designed to improve performance on a single training environment (e.g., NoisyNet, ez-greedy, bootstrapped DQN etc.) are insufficient when applied in such settings. This motivated the design of EDE, a new exploration method that significantly improves performance in new environments. While it is true that EDE has strong empirical performance (especially for a value-based method), we think the algorithmic contribution is secondary to the conceptual message as algorithms can always be improved. Since most existing methods for Procgen do not target exploration explicitly, we think they would benefit from better exploration. We hope our work is a first step towards designing exploration methods that improve generalization in RL and that it will inspire more research on better understanding the relationship between exploration and generalization.


> Choice of baselines (reviewers PttU and dHNU)

Exploration is a longstanding problem in RL and it is infeasible for us to evaluate all possible methods in the literature. We believe that the baselines and ablations we use in the paper are extensive and representative of general-purpose exploration methods in the literature. The estimated time for all Procgen experiments (without considering hyperparameter tuning) is more than **250 GPU days** and covers **5 popular baselines + 8 ablations**. We believe this ensures that the conclusions from our experiments are reliable.

Most of the baselines we chose are general-purpose exploration methods that aim to accelerate learning rather than solve an environment with extremely sparse reward (which is not the setting we consider here and would require additional techniques). They are widely cited and distinct from each other (e.g., bootstrapped DQN is cited 1222 times and NoisyNet is cited 568 times). They follow the principle of "optimism in the face of uncertainty", trading off return maximization with uncertainty. We will refer to them as optimism-based exploration although the exact implementation can vary. Since EDE follows the same principle to explore, we believe these are the most relevant and informative baselines to compare against.

Several reviewers mentioned intrinsic reward methods which we assume refer to methods such as ICM [2] or RND [3]. As we explained in footnote 4, [1] finds that *existing intrinsic reward methods are no better and much less sample-efficient than optimism-based exploration methods such as NoisyNet on Atari*, except for the environments with extremely sparse reward. Note that Procgen and Crafter don't fall into this category since they generally have dense rewards and not-so-long episodes. Thus, we think it is unlikely that intrinsic reward methods would outperform our baselines (e.g., NoisyNet), so we did not include them due to the limited computation budget. However, we believe that a better understanding of how intrinsic rewards influence generalization in RL is an important and interesting research question that would be better addressed in future work.


**References**

[1] On bonus based exploration methods in the arcade learning environment. Taiga et al. 2021.

[2] Curiosity-driven Exploration by Self-supervised Prediction. Pathak et al. 2017.

[3] Exploration by Random Network Distillation. Burda et al. 2018.

**Conclusion**

We thank you again for your feedback and support of the paper. We hope our clarifications have adequately addressed all your concerns but let us know if you have any remaining questions.

---

> ### Comment · Area_Chair_fJEq · 2023-08-18
>
> Dear authors,
>
> Thanks for your rebuttal. Unfortunately, not all reviewers have responded to your rebuttal yet. I just wanted to let you know that I'll keep trying to get the to reply to your rebuttal. In any case, the points raised in the rebuttal will be taken into account in the private reviewer-AC discussion and in the final decision making.
>
> -The AC

---

### Decision · Program_Chairs · 2023-09-21

**Decision:**

Accept (poster)

**Comment:**

The paper addresses the role of exploration in generalization for cMDP solution strategies. The paper proposes a ensemble-based method using quantized Q-funcitons to quantify epistemic and aleatoric uncertainty. The inferred epistemic uncertainty is then used to guide exploration. The method is tested on two cMDP enviornments that serve as benchmarks for generalization: Crafter and ProcGen, and outperforms the considered baselines in these environment.

There was a extended discussion on these papers, both between the authors and the reviewers and amongst the reviewers themselves.

The reviewers mostly agreed that the paper was well written, provided insight on a significant problem, and contained original ideas. Several reviewers also noted that from the results it is hard to understand the differences in exploration behavior.

There was discussion on the significance of the contribution of the paper. The contribution was rated as “2, fair” by 3 of the five reviewers. The novelty of the paper seems to be in two main aspects:

1. Introducing a new methods, composed of elements of two known approaches (quantile learning and ensemble methods) to improve generalization
2. making the conceptual connection between exploration and generalization.

The results presented by the authors on two domains are quite convincing compared to the considered benchmarks. However, the experiments could have been more convincing in several aspects, such as (1) more directly measuring exploration (2) considering more environments (3) more directly measuring generalization. As is, the experiments measure policy performance, which makes it hard to exactly understand whether (1) the performance is explained by other factors such as tuning (2) the performance is explained by improved exploration alone (3) the performance improvement is due to improved generalization.

Another point of discussion was the relationship between the example in section 3 and the method development in section 4. The authors clarified in the discussion and rebuttal that the example in section 3 is a motivating example and not part of the core contribution, and most reviewers found the example helpful if not perhaps ideally suited.